# S-adenosylmethionine synthases specify distinct H3K4me3 populations and gene expression patterns during heat stress

Adwait A Godbole[1], Sneha Gopalan[2,3], Thien-Kim Nguyen[1], Alexander L Munden[1], Dominique S Lui[1], Matthew J Fanelli[1], Paula Vo[1], Caroline A Lewis[1], Jessica B Spinelli[1,2], Thomas G Fazzio[2,3], Amy K Walker[1,3]*

[1]Program in Molecular Medicine, UMASS Chan Medical School, Worcester, United States; [2]Cancer Center, UMASS Chan Medical School, Worcester, United States; [3]Department of Molecular, Cell, and Cancer Biology, UMASS Chan Medical School, Worcester, United States

**Abstract** Methylation is a widely occurring modification that requires the methyl donor S-adenosylmethionine (SAM) and acts in regulation of gene expression and other processes. SAM is synthesized from methionine, which is imported or generated through the 1-carbon cycle (1 CC). Alterations in 1 CC function have clear effects on lifespan and stress responses, but the wide distribution of this modification has made identification of specific mechanistic links difficult. Exploiting a dynamic stress-induced transcription model, we find that two SAM synthases in *Caenorhabditis elegans*, SAMS-1 and SAMS-4, contribute differently to modification of H3K4me3, gene expression and survival. We find that *sams-4* enhances H3K4me3 in heat shocked animals lacking *sams-1*, however, *sams-1* cannot compensate for *sams-4*, which is required to survive heat stress. This suggests that the regulatory functions of SAM depend on its enzymatic source and that provisioning of SAM may be an important regulatory step linking 1 CC function to phenotypes in aging and stress.

*For correspondence:
amy.walker@umassmed.edu

Competing interest: The authors declare that no competing interests exist.

## Editor's evaluation

The manuscript by Godbole et al. proposes a novel mechanism by which different S-adenosylmethionine (SAM) synthase enzymes exhibit specificity towards target sequences, establishing a layer of control over H3K4 trimethylation (H3K4me3). The authors demonstrate that the loss of two SAMs (*sams-1* and *sams-4*) differentially impacts stress response phenotypes, histone methylation, and gene expression profiles. This work suggests a role of enzyme provisioning in selecting specific targets for epigenetic modification.

## Introduction

The 1-Carbon cycle (1 CC) is a group of interconnected pathways that link essential nutrients such as methionine, folate, and vitamin B12 to the production of nucleotides, glutathione, and S-adenosylmethionine (SAM), the major methyl donor (*Ducker and Rabinowitz, 2017*; *Figure 1A*). SAM is important for the production of polyamines and phosphatidylcholine (PC), a methylated phospholipid, and is also essential for the methylation of RNA, DNA and proteins such as histones (*Mato et al., 2008*). Thus, 1 CC connects nutrients with the production of a key cellular regulator of epigenetic function, SAM.

Alterations in 1 CC function can cause a variety of defects (*Ducker and Rabinowitz, 2017*), including intriguing connections between this cycle, stress responses and aging. Lifespan lengthens in yeast, *C. elegans*, *Drosophila* and rodent models when methionine is restricted, genes in the methionine-SAM (Met-SAM) cycle are mutated, or polyamines are supplemented (*Parkhitko et al., 2019*). While multiple aspects of 1 CC function could affect aging, the Met-SAM cycle has particularly strong links. For example, a *C. elegans* SAM synthase, *sams-1*, was identified in a screen for long-lived animals (*Hansen et al., 2005*) and multiple SAM-utilizing histone methyltransferases are also implicated as aging regulators (*Han and Brunet, 2012*; *Greer et al., 2010*; *Han et al., 2017*). Of bioactive molecules, SAM is second only to ATP in cellular abundance (*Ye and Tu, 2018*), which raises the question of how such an abundant metabolite can exert specific phenotypic effects. Strikingly, studies in multiple organisms from a variety of labs have shown that reduction in SAM levels preferentially affects H3K4me3 levels (*Mentch et al., 2015*; *Shyh-Chang et al., 2013*; *Kraus et al., 2014*; *Ding et al., 2015*). However, changes in SAM production may affect other histone modifications as well. For example, the Gasser lab showed that *sams-1* and *sams-3* have distinct roles in heterochromatin formation, which involves H3K9me3 (*Towbin et al., 2012*) A yeast SAM synthase has also been shown to act as part of the SESAME histone modification complex (*Li et al., 2015*) or to cooperate with the SIN3 repressor (*Liu and Pile, 2017*). In addition, most eukaryotes have more than one SAM synthase, which could allow partitioning of enzyme output by developmental stage, tissue type or cellular process and underlie specific phenotypic effects. Indeed, in budding yeast, SAM1 and SAM2 are co-expressed but regulated by different metabolic events, have distinct posttranslational modifications, and act differently in phenotypes such as genome stability (*Hoffert et al., 2019*). The two SAM synthases present in mammals are expressed in distinct tissues: MAT2A is present throughout development and in most adult tissues, whereas MAT1A is specific to adult liver (*Maldonado et al., 2018*). MAT2A may be present in distinct regulatory conformations with its partner MAT2B (*Maldonado et al., 2018*). However, the distinct molecular mechanisms impacted by these synthases are less clear. Studies exploring specificity of metazoan SAM synthase function have been difficult, as MAT1A expression decreases ex vivo and MAT2A is essential for cell viability (*Mato et al., 2002*). Finally, the high methionine content of traditional cell culture media has limited functional studies (*Sullivan et al., 2021*).

We have explored SAM synthase function in *C. elegans*, where the gene family has undergone an expansion. In *C. elegans*, genetic and molecular assays allow separation of SAM synthase expression and function in vivo. Furthermore, no single SAM synthase is required for survival in normal laboratory conditions or diets. *sams-1* and the highly similar *sams-3*/*sams-4* are expressed in adult animals, whereas *sams-5* is present at low levels in adults and *sams-2* is a pseudogene (*Harris et al., 2020*). We previously found that *sams-1* had multiple distinct functions, contributing to PC pools and stimulating lipid synthesis through a feedback loop involving *sbp-1*/SREBP-1 (*Walker et al., 2011*) as well as regulating global H3K4me3 levels in intestinal nuclei *Ding et al., 2015*. Our studies also showed that loss of *sams-1* produced different phenotypes in bacterial or heat stress. While *sams-1* was necessary for pathogen challenge, promoter H3K4me3 and expression of immune genes, animals surprisingly survived better during heat shock when they lacked *sams-1* (*Ding et al., 2015*). Because heat shocked animals require the H3K4me3 methyltransferase *set-16/MLL* for survival, we hypothesized that SAM from a different source may be important for histone methylation and survival in the heat shock response. Here, we find that SAM source impacts the functional outputs of methylation. While the SAM and the 1 CC are well associated with regulation of lifespan and stress responses, direct molecular connections have been difficult to discover. Mechanisms controlling provisioning of SAM, therefore, could provide a critical level of regulation in these processes. We show that *sams-1* and *sams-4* differentially affect different populations of histone methylation and thus gene expression in the heat shock response, and that their loss results in opposing phenotypes. Our study demonstrates that SAM synthases have a critical impact on distinct methylation targets and phenotypes associated with the stress response. Thus, defining the specificity of SAM synthases may provide a method to identify from broad effects methylation events that are specific phenotypic drivers.

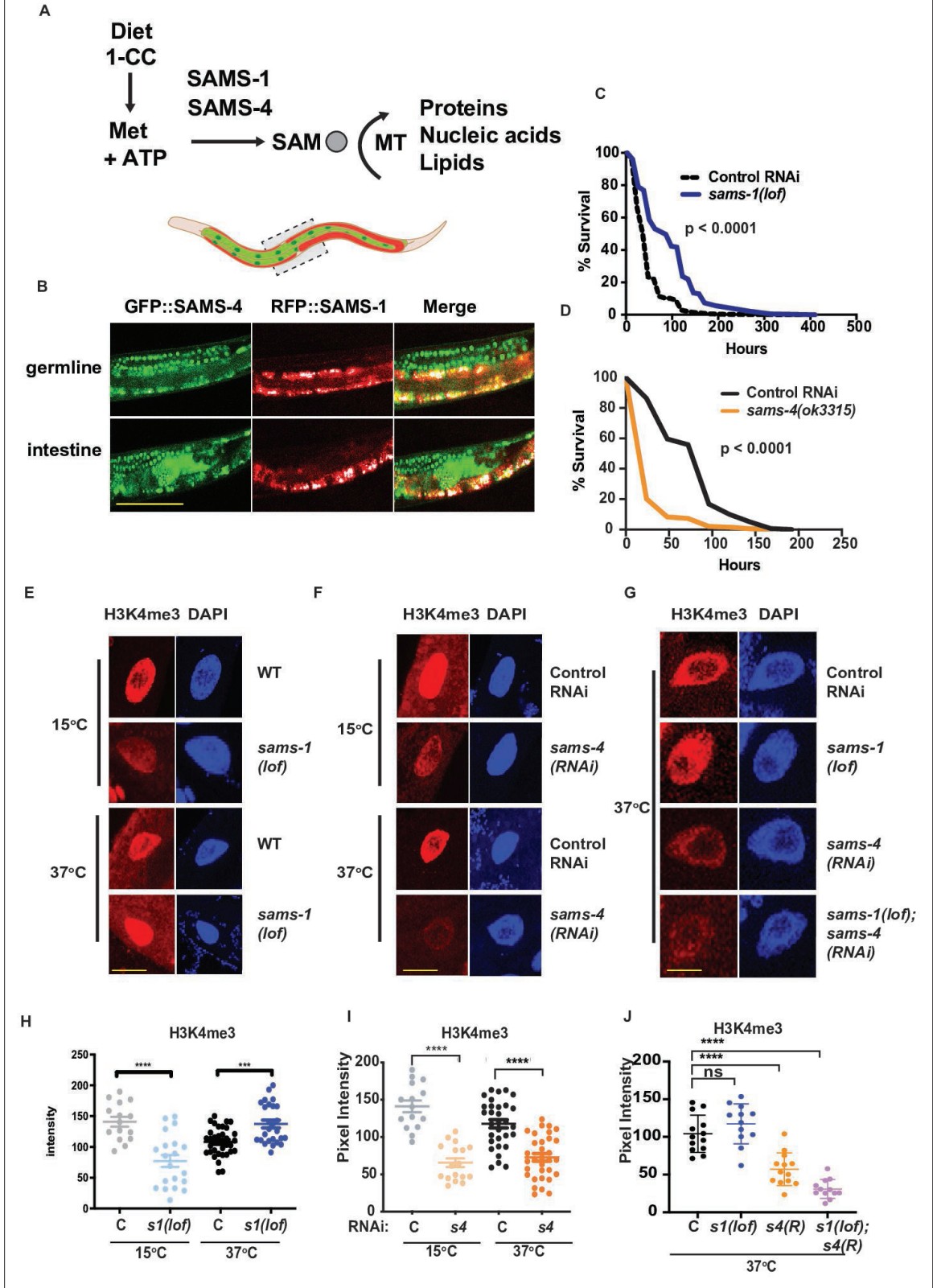

**Figure 1.** acquisition of H3K4me3 in heat-shocked animals. (**A**) Methionine intake through diet enters the 1 carbon cycle and is used by SAM synthases for the synthesis of SAM which is used by methyltransferases to add methyl moieties to proteins, nucleic acids and lipids. (**B**) Representative confocal images of animals co-expressing RFP::SAMS-1and GFP::SAMS-4 in the germline and intestine. Scale bar represents 50 microns. Kaplan-Meier survival plots of *sams-1(lof)* (**C**) or *sams-4(ok3315)* (**D**) following heat shock. Statistical significance is shown by Log-rank test. Each graph represents the compiled

*Figure 1 continued on next page*

*Figure 1 continued*

data from three biologically independent repeats; data is compiled in ***Supplementary file 2***. Representative immunofluorescence images of intestinal nuclei stained with H3K4me3-specific antibody and quantification in *sams-1(lof)* animals (**E, H**), *sams-4(RNAi)* (**F, I**) or in *sams-1(lof); sams-4(RNAi)* animals (**G, J**). *sams-3* may also be targeted; see also (***Figure 3E***). Scale bar represents 25 microns. Error bars show average and standard deviation. Statistical significance was calculated using unpaired Student's t-test. ns = not significant, ****=p < 0.0001, ***=p < 0.001. Graph represents compiled data from three biologically independent repeats per condition with each point representing a single animal.

The online version of this article includes the following figure supplement(s) for figure 1:

**Figure supplement 1.** Expression patterns of SAM synthases in adult *C. elegans*.

**Figure supplement 2.** Distinct patterns of gene expression after *sams-1 or sams-4* RNAi in basal conditions.

**Figure supplement 3.** *sams-4* is important for survival and H3K4me3 in *sams-1* animals after heat shock.

## Results

### *sams-1* and *sams-4* have overlapping and distinct expression patterns and gene regulatory effects

Animals respond to stress by activating specialized protective gene expression programs (*de Nadal et al., 2011*). While these programs depend on specific signaling and transcriptional activators, they may also be impacted by histone methylation and the production of SAM. For example, we found that *C. elegans* lacking *sams-1* die rapidly on pathogenic bacteria, have low global H3K4me3 and fail to upregulate immune response genes (*Ding et al., 2015*). In contrast, heat shocked animals survive better without *sams-1* (*Ding et al., 2018*). *sams-1(RNAi)* animals induced heat shock genes to normal levels and acquired additional changes in the transcriptome, including downregulation of many metabolic genes. However, the H3K4me3 methyltransferase *set-16*/MLL was essential for survival (*Ding et al., 2018*), suggesting that methylation was required. We hypothesized that other SAM synthases could play an important role in mediating survival during heat shock (***Figure 1A***).

In order to test these hypotheses, we first compared expression of each synthase, SAM levels and gene expression after RNAi in adult unstressed animals. ModEncode data *Gerstein et al., 2010* from young adult animals shows that in young adult levels, *sams-1* is expressed at the highest levels, comparable to the metabolic enzyme GAPDH (*gpdh-1*) (***Figure 1***, ***Figure 1—figure supplement 1A***). *sams-3* and *sams-4* are expressed at lower levels, but comparable to other enzymes of the 1-Carbon cycle such as *metr-1*, whereas *sams-5* is minimally expressed (***Figure 1—figure supplement 1A***). In order to determine the tissue-specific patterns of the SAM synthases expressed in adult animals, we obtained strains where each protein was tagged with RFP, GFP or mKate, via CRISPR (***Figure 1B***, ***Figure 1—figure supplement 1B, C***). RFP::*sams-1* and GFP::*sams-4* animals were also crossed to allow visualize expression of both synthases (***Figure 1B***). RFP::SAMS-1fluorescence was evident in much of the adult animal, including intestine, hypodermis and cells in the head (***Figure 1B***, ***Figure 1—figure supplement 1B***), in line with mRNA expression patterns derived from tissue-specific RNA seq (*Kaletsky et al., 2018*). However, RFP::SAMS-1was not present in the germline, which did express GFP::SAMS-4 and SAMS-3::mKate (***Figure 1B***, ***Figure 1—figure supplement 1C***). GFP::SAMS-4 and SAMS-3::mKate was also present in intestinal and hypodermal cells (***Figure 1B***, ***Figure 1—figure supplement 1C***), demonstrating that these tissues, which are major contributors to the stress response (*McGhee, 2007*) contain each of these SAM synthases. *sams-3* and *sams-4* are expressed bidirectionally from the same promoter and share 95% sequence identity at the nucleotide level thus RNAi targeting is likely to affect both genes. Indeed SAMS-3::mKate and GFP::SAMS-4 were reduced after either RNAi (***Figure 1—figure supplement 1C***). Next, we used mass spectrometry to compare SAM levels after *sams-3* and *sams-4* RNAi and found that like *sams-1* (*Ding et al., 2015*; *Walker et al., 2011*), reduction in any synthase significantly reduced but did not eliminate SAM (***Figure 1—figure supplement 1D***).

In order to compare gene expression after RNA of each SAM synthase in basal conditions, we used RNA sequencing (RNAseq). Principal component analysis showed that *sams-1(RNAi)* and *sams-5* formed distinct clusters on the first two principal components; however, *sams-3* and *sams-4* were overlapping (***Figure 1—figure supplement 2A***; ***Supplementary file 1***: Tabs A-C). About half of the genes upregulated after *sams-4* knockdown also increased in *sams-1(RNAi)* animals (***Figure 1—figure supplement 2B***). To determine if genes related to distinct biological processes were present, we compared genes upregulated after *sams-1* RNAi (*Ding et al., 2018*) with those changing in *sams-4*

RNAi with WormCat (*Holdorf et al., 2020*), which provides enrichment scores for three category levels (Cat1, Cat2, Cat3) for broad to more specific comparisons. WormCat finds that gene function categories at the Cat1 and Cat 2 level, such as METABOLISM: Lipid (*Figure 1—figure supplement 2C*) or STRESS RESPONSE: Pathogen (*Figure 1—figure supplement 2D–F*), are enriched at lower levels and contain different genes in *sams-4(RNAi)* animals (*Supplementary file 1*: Tabs D-F). Notably, *fat-7* and other lipid synthesis genes that respond to low PC in *sams-1* animals are not upregulated after *sams-4(RNAi)* (*Supplementary file 1*:**Tab:B**). These findings strengthen the idea that these SAM synthases could have distinct functions.

## Opposing roles and requirements for *sams-1* and *sams-4* in the heat shock response

In order to determine if other SAM synthases expressed in adult animals contributed to survival in heat shock (*Figure 1—figure supplement 3A*), we compared the heat shock survival phenotypes of *C. elegans* with deletions in *sams-1*, *sams-3* and *sams-4* to avoid effects of co-targeting by RNAi. *sams-1(ok3033)* has a deletion covering the majority of the open reading frame and extracts from these animals lack SAMS-1 protein in immunoblots (*Ding et al., 2015*); therefore, we refer to this allele as *sams-1(lof)*. *sams-4(ok3315)* animals have a deletion that removes around a third of the open-reading frame. Strikingly, *sams-4(ok3315)* mutants had the opposite phenotype from *sams-1(lof)*, and died rapidly after heat shock (*Figure 1C and D*, *Supplementary file 2*:Tabs B, C). *sams-3(2932)* harbors a deletion removing most of the ORF, but in contrast to *sams-4* and *sams-1,* is indistinguishable from wild type animals in a heat shock response (*Figure 1—figure supplement 3B*). Although *sams-3* may be co-targeted in RNAi experiments, we will refer solely to *sams-4* in our discussion because it has the most direct link to the heat shock phenotypes. Finally, *sams-4(RNAi)* phenotypes in the heat stress response were not linked to a general failure to thrive, as *sams-4(RNAi)* animals under basal conditions had modestly enhanced lifespan (*Figure 1—figure supplement 3C*; *Supplementary file 2*: Tab A).

Next, we used immunostaining to compare global levels of H3K4me3 in *sams-1* and *sams-4* RNAi nuclei during heat shock. In contrast to the reduction in H3K4me3 in basal conditions in *sams-1(lof)*, *sams-4(ok 3315)* or RNAi animals (*Figure 1E–F–*), we detected robust levels of H3K4me3 in *sams-1(lof)* nuclei after heat shock (2 hr at 37 °C) (*Figure 1E and H*), suggesting that *sams-1*-independent mechanisms act on H3K4me3 during heat shock. These increases in H3K4me3 did not appear in heat shocked *sams-4(RNAi)* intestinal nuclei *Figure 1F, I*, raising the possibility that *sams-4* contributed to the effects in *sams-1(lof)* animals. Next, we wanted to test effects of reducing both *sams-1* and *sams-4* levels on H3K4me3 during heat shock. Loss of multiple SAM synthases reduces viability in *C. elegansTowbin et al., 2012*. In order to circumvent this, we used dietary choline to rescue PC synthesis and growth of *sams-1(RNAi)* or *(lof)* animals during development (*Ding et al., 2015*; *Walker et al., 2011*). *sams-1(lof); sams-4(RNAi)* animals were raised on choline until the L4 stage, then moved to normal media for 16 hr before heat shock. Immunostaining of *sams-1(lof); sams-4(RNAi)* intestines showed that *sams-4* is required for the H3K4me3 in heat shocked *sams-1(lof)* nuclei (*Figure 1G and J*). These results were identical when we used RNAi to reduce *sams-1* in *sams-4(ok3315)* animals (*Figure 1—figure supplement 3D*). We also asked if *sams-4* was necessary for the increased survival of *sams-1* animals after heat shock and found that the survival advantage in *sams-1(RNAi)* was decreased in *sams-4(ok3315)* animals (*Figure 1—figure supplement 3E*). These results suggest that H3K4me3 may be remodeled during heat shock with SAM from distinct synthases and that *sams-4*-dependent methylation is critical for survival. Previously, it was shown that H3K4me3 deposition is independent of *sams-4* in embryonic nuclei (*Towbin et al., 2012*), however, our finding that it is broadly decreased in *sams-4(RNAi)* intestinal nuclei suggests it may have important roles in H3K4 methylation in adults.

Increases in H3K4me3 have also been shown to occur in budding yeast when blocks in phospholipid synthesis relieve a drain on SAM and increase levels (*Ye et al., 2017*), which we have confirmed in *C. elegans* (*Ding et al., 2018*). In order to determine if SAM levels could explain differences in H3K4me3 in *sams-1* and *sams-4* animals during heat shock, we used targeted LC/MS to compare SAM, it's precursor methionine and S-adenosylhomocysteine (SAH), the product after methyl transfer, before and after heat shock. As in our previous assays, SAM decreased significantly after *sams-1* or *sams-4(RNAi)* in basal conditions (*Figure 1—figure supplement 3F*), whereas SAM levels increased in each population as *sams-1* or *sams-4* animals were shifted to 37 °C for 2 hr (*Figure 1—figure supplement 3F*). Levels of methionine and SAH also decreased when comparing control, *sams-1* or

sams-4(RNAi) animals in basal vs heat-shocked conditions (*Figure 1—figure supplement 3G, H*), consistent with increased production and utilization of SAM. The increase in SAM in heat-shocked animals is consistent with our data showing the contribution of SAMS-4 to H3K4me3 and survival in heat-shocked *sams-1* animals; however, a reduction in demand for SAM if other metabolic processes are reduced after heat shock could also contribute. Finally, levels of SAM in heat-shocked *sams-4(RNAi)* animals also rise to levels comparable to control animals at basal temperatures; however, H3K4me3 remains low in these conditions.

## Histone methyltransferase and histone demethylation machinery have modest, separable effects on *sams* mutant heat shock phenotypes

SAM is necessary for histone methylation; however, histone methylation dynamics are also influenced by methyltransferase (KMT) or demethylase (KDMT) activity (*Bannister and Kouzarides, 2011*). Therefore, changes in histone methylation dynamics could also impact H3K4me3 patterns during heat shock. H3K4me3 is catalyzed by multiple versions of the COMPASS complex, which each consist of one of several SET domain histone methyltransferases and several shared accessory subunits (*Shilatifard, 2012*). In mammals, seven methyltransferases in the SET1, MLL or THX groups can methylate H3K4. *C. elegans* contain single orthologs from two of these groups: *set-2/*SET1 and *set-16/*MLL, respectively, with roles in embryonic development (*Li and Kelly, 2011*; *Xiao et al., 2011*; *Wenzel et al., 2011*), lipid accumulation and transgenerational inheritance (*Greer et al., 2010*; *Han et al., 2017*). In adult *C. elegans*, *set-2* RNAi results in extensive loss of H3K4me3 in intestinal nuclei and although *set-16(RNAi)* causes an intermediate reduction in bulk H3K4me3 levels, it has a broader requirement for survival during stress (*Ding et al., 2018*). Because specificity for H3K4 mono, di or trimethylation has not been verified on a genome-wide scale for KDMTs, we examined multiple members of the H3K4 KDM family.

In order to determine if KMTs or KDMT dynamics played a role in the change of H3K4me3 during heat shock, we used RNAi to deplete them in *sams-1(lof)* or *sams-4(ok3315)* animals and measured survival after heat shock and intestinal H3K4me3 levels. RNAi of *set-2/SET1* (*Figure 2A*) or *set-16/MLL* (*Figure 2B*) increased survival in *sams-1(lof)* animals after heat shock (also *Supplementary file 2*:Tabs:C, E) and did not limit heat shock-induced H3K4me3 in *sams-1(RNAi)* nuclei (*Figure 2D and E*; GH). RNAi of two KDMTs, *rbr-2* (*Figure 2C*) and *spr-5* (*Figure 2—figure supplement 1A*) had opposite effects from the KMTs, moderately reducing survival (*Supplementary file 2*: Tab F), whereas *amx-1* and *lsd-1* had no effect (*Figure 2—figure supplement 1B, C*; *Supplementary file 2*: Tabs I, J). RNAi of *set-2* or *set-16* had slight, but statistically significant effects, increasing survival of *sams-4(ok3315)* animals (*Figure 2—figure supplement 1D and E*; *Supplementary file 2*: Tabs G, H). However, survival was still significantly below controls in *sams-4(ok3315)* with or without the KMT RNAi. Taken together, this suggests that *set-2* and *set-16* may act redundantly in the deposition of H3K4me3 after heat shock and are important to survival in *sams-1(lof)* animals. Furthermore, our data illustrate that the context is critical for understanding role of SAM and H3K4me3 in stress; *sams-4* and *set-16* are generally required for survival after heat shock, but loss of either H3K4 KTM enhances survival in *sams-1(lof)* animals.

## Distinct patterns of H3K4me3 and gene expression in *sams-1(RNAi)* versus *sams-4(RNAi)* animals during heat shock

H3K4me3 is a prevalent modification enriched near the transcription start sites (TSSs) of actively expressed genes (*Eissenberg and Shilatifard, 2010*). Differing global patterns of H3K4me3 in *sams-1(RNAi)* and *sams-4(RNAi)* nuclei suggest this histone modification at specific sites could also be distinct. In order to identify loci that might link H3K4me3 to these phenotypes, we used CUT&Tag, (Cleavage Under Targets and Tagmentation, C&T) (*Kaya-Okur et al., 2019*), to determine genome-wide H3K4me3 levels in Control RNAi, *sams-1* and *sams-4(RNAi)* in basal (15 °C) and after heat shock (37 °C/2 hr) from two biologically independent replicates along with no antibody controls. C&T is uniquely suited to the small sample sizes available from these stressed populations. In this approach, a proteinA-Tn5 transposase fusion protein binds to the target antibody in native chromatin and DNA libraries corresponding to antibody binding sites are generated after transposase activation. After sequencing of libraries, we used the HOMER analysis suite *Heinz et al., 2010* to analyze reads mapped to the *C. elegans* genome and called peaks using ChIPSeqAnno *Zhu et al., 2010* for

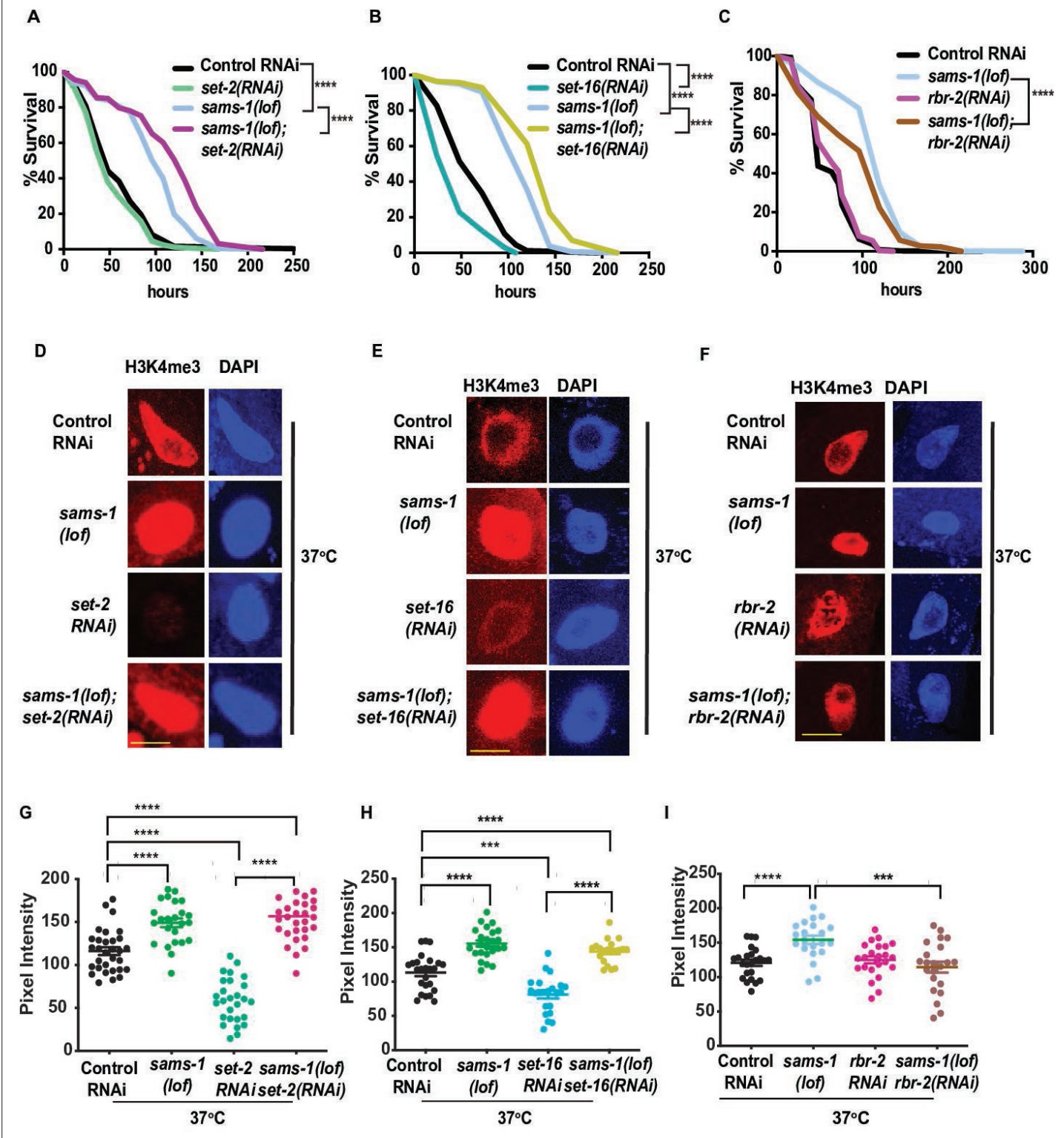

**Figure 2.** H3K4me3 demethylases modulate SAM synthase phenotypes. Kaplan-Meier plots of survival assays comparing basal and heat shocked wild type (*N2*) or *sams-1(lof)* animals grown on RNAi for the histone methyltransferases *set-2* (**A**) and *set-16* (**B**), or demethylases *rbr-2* (**C**) and *spr-5* (**D**).Scale bar is 25 microns. Heat shock survival assays for *sams-4(ok3315)* animals exposed to *set-2* or *set-16* RNAi are shown in (**E, F**). Statistical significance is shown by Log-rank test. Each graph represents compiled data from 3 biologically independent repeats. Data for each replicate is compiled in *Supplementary file 2*. Black bars show mean and standard deviation. Statistical significance is determined by Student T test.

The online version of this article includes the following figure supplement(s) for figure 2:

*Figure 2 continued on next page*

*Figure 2 continued*

**Figure supplement 1.** H3K4me3 demethylases modulate SAM synthase phenotypes.

more detailed peak annotation. Bar plots from ChIPSeqAnno annotations and TSS plots generated with HOMER show robust mapping of H3K4me3 to promoter-TSS regions, validating this approach (*Figure 3A*; *Supplementary file 3*: Tabs A-F). While promoter-TSS regions were the largest feature in each sample, heat shocked *sams-4(RNAi)* animals had fewer overall peaks (*Figure 3A*). Correlation plots also show strong similarity between replicates (*Figure 3—figure supplement 1A*). Because C&T has not been extensively used in *C. elegans*, we compared data from basal conditions in our study to three previously published ChIP-Seq data sets (*Ho et al., 2014*; *Pu et al., 2015*; *Wan et al., 2022*). We compared our C&T data from wild type young adult animals grown at 15 °C on control RNAi food (HT115) against ModEncode (L3 animals), *glp-1(e2141)* mutants from *Pu et al., 2018* and wild type adults grown at 20 °C on OP50 bacteria from *Wan et al., 2022* by computing a pair-wise Pearson correlation. We found our C&T clustered most closely with the ChiPSeq from wild type animals in Wan et al., along with one of the modEndode replicates (*Figure 3—figure supplement 1B*) with moderate correlation scores. Both our C&T data and the Wan ChiPseq data correlated poorly with the Pu et al. ChIP seq, which is likely due to the lack of germline nuclei in these animals. The moderate correlation between our data and ChiP seq from Wan et al may be due to differences in growth temperature and bacterial diet. As a part of our quality control, we visually inspected browser tracks around the *pcaf-1* gene, which is a long gene and has been used by our labs and others as a positive control for H3K4me3 localization in the five prime regions (*Ding et al., 2015*; *Xiao et al., 2011*). H3K4me3 peaks are prominent upstream of the transcript as expected and the no antibody libraries showed few reads (*Figure 3—figure supplement 1C*).

Next, we compared TSS distributions and examined overlap between H3K4me3 peaks in Control RNAi animals in basal and heat shock conditions and found moderate reductions occurred with heat shock (*Figure 3B*). Around 20–30% of peaks were specific to at either at basal (15 °C) vs. heat shock (37 °C) temperature (*Figure 3C*), suggesting that H3K4me3 could be remodeled upon heat shock in *C. elegans*. TSS enrichment of H3K4me3 was sharply reduced in both *sams-1* and *sams-4* samples at 15 °C; however, this difference was less marked in heat-shocked animals, in line with lower TSS localization in Control animals (*Figure 3D and E*). While aggregate TSS enrichment for H3K4me3 was similar for *sams-1* and *sams-4* RNAi animals, this analysis could miss distinct sets of H3K4me3 marked genes in each condition. Indeed, Control, *sams-1* and *sams-4(RNAi)* animals each showed 500–1000 specific peaks in basal conditions, with moderate increases in these numbers after heat shock (*Figure 3D and E*). As H3K4me3 is a widely occurring modification, we hypothesized that we might better understand potential SAM synthase-specific requirements if we focused on peaks that change in the Control RNAi heat shock response and asked how they are affected by loss of *sams-1* or *sams-4*. We used two different methods for comparing potential SAM synthase requirements for H3K4me3 in the heat shock response. First, we used differential peak calling ChIPPeakAnno *Zhu et al., 2010* followed by WormCat category enrichment to determine the classes of genes which might be affected (*Figure 3—figure supplement 2A–F*; *Supplementary file 3*; Tabs G-I). Peaks present in both basal and heat shocked conditions were enriched for genes in the METABOLISM category (including Lipid: phospholipid, sphingolipid, sterol and lipid binding, along with mitochondrial genes) as well as in core function categories such as those involved in trafficking, DNA or mRNA functions (*Figure 3F*, *Figure 3—figure supplement 2D–E*; *Supplementary file 3*: Tabs G-I). There was no significant category enrichment specific to 15 °C animals, but after heat shock, Control RNAi animals gain enrichment in peaks at the Category 1 level in PROTEOSOME PROTEOLYSIS (*Figure 3F*). This enrichment was driven by increases in H3K4me3 at E3: Fbox genes (*Figure 3—figure supplement 2A, B*; *Supplementary file 3*:Tab A,B), which could be important for eliminating mis-folded proteins during heat shock. Comparison of peaks differentially present in *sams-1* and *sams-4* RNAi animals showed that only *sams-1(RNAi)* exhibited a similar enrichment to Control RNAi in the PROTEOLYSIS PROTEOSOME category (*Figure 3G*, *Figure 3—figure supplement 2C, D*), which could help explain the reduced survival of *sams-4(RNAi)* animals relative to *sams-1(RNAi)* animals. *sams-1* RNAi animals also gained enriched peaks in a wide range of gene categories within METABOLISM, whereas *sams-4(RNAi)* enriched peaks in these categories were more limited (*Figure 3—figure supplement 2C–F*).

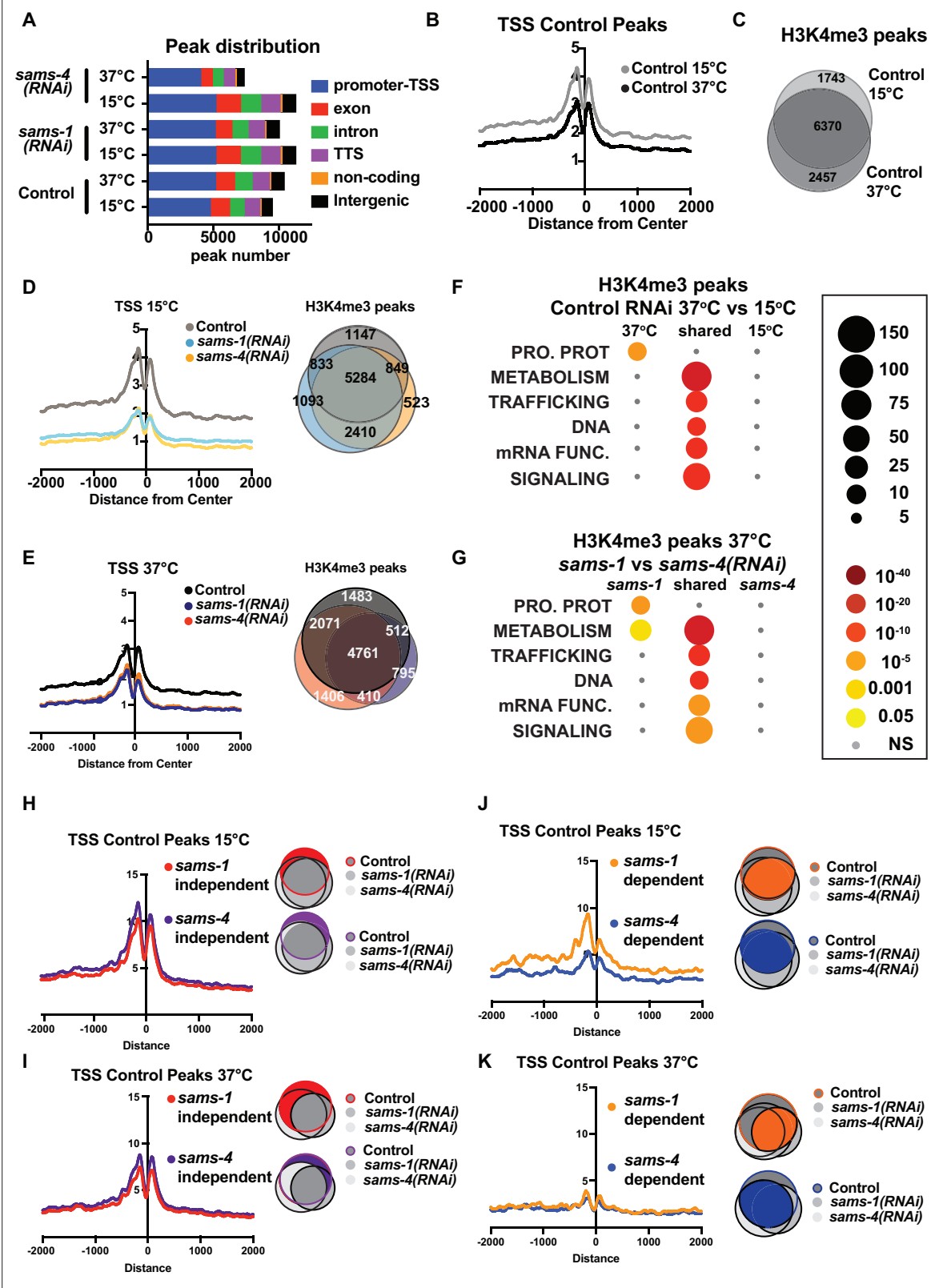

**Figure 3.** H3K4me3 modifying enzymes modulate SAM synthase phenotypes. (**A**) Bar graph showing the distribution of the enrichment of H3K4me3 over different genomic loci in animals fed control RNAi, *sams-1(RNAi)* or *sams-4(RNAi)* at 15°C and 37°C. (**B**) Aggregation plots showing TSS enrichment in the H3K4me3 peaks identified in animals fed control RNAi at 15°C and 37°C. The Y axis on TSS plots shows Peaks per base pair of gene. (**C**) Venn diagram comparing the overlap in the H3K4me3 peaks identified in animals fed control RNAi at 15°C and 37°C. (**D**) Aggregation plots showing TSS

*Figure 3 continued on next page*

*Figure 3 continued*

enrichment in the H3K4me3 peaks identified in animals fed control RNAi or *sams-1(RNAi)* or *sams-4(RNAi)* at 15 °C and Venn diagram comparing the overlap in the H3K4me3 peaks identified in animals fed control RNAi or *sams-1(RNAi)* or *sams-4(RNAi)* at 15 °C. (**E**) Aggregation plots showing TSS enrichment in the H3K4me3 peaks identified in animals fed control RNAi or *sams-1(RNAi)* or *sams-4(RNAi)* at 15 °C and Venn diagram comparing the overlap in the H3K4me3 peaks identified in animals fed control RNAi or *sams-1(RNAi)* or *sams-4(RNAi)* at 37 °C. (**F**) Bubble chart showing enriched gene categories in differential peaks as determined by WormCat in animals fed control RNAi at 15 °C only, 37 °C only and common between 15°C and 37°C (**G**) or *sams-1(RNAi)* and *sams-4(RNAi)* at 37 °C. Aggregation plots showing TSS enrichment of Control peaks that did not change after *sams-1(RNAi)* and *sams-4(RNAi)* (independent) (**H**) 15 °C or (**I**) 37 °C. Shaded areas in the Venn diagrams indicate the population of genes used for plotting the TSS enrichment plots. Aggregation plots showing TSS enrichment of Control peaks that were dependent on *sams-1(RNAi)* or *sams-4(RNAi)* (**J**) 15 °C or (**K**) 37 °C. Shaded areas in the Venn diagrams indicate the population of genes used for plotting the TSS enrichment plots.

The online version of this article includes the following figure supplement(s) for figure 3:

**Figure supplement 1.** H3K4me3 C&T correlation with published H3K4me3 ChipSeq data.

**Figure supplement 2.** Distinct H3K4me3 patterns after heat shock in *sams-1* and *sams-4* RNAi animals.

**Figure supplement 3.** SAM synthase-specific patterns H3K4me3 in germline nuclei.

Thus, loss of *sams-1* or *sams-4* differentially affects H3K4me3 peaks within functional gene classes that also change in the heat shock response.

Next, we hypothesized that H3K4me3 at peaks in Control RNAi animals might reflect multiple differently regulated populations, some which are linked to SAM synthase function and others that are regulated at other levels. In order to test this, we divided peaks in Control animals at 15 °C or 37 °C into those that did not change after SAM synthase RNAi (*sams-1* or *sams-4* independent peaks) or those that were dependent on *sams-1* or *sams-4* and examined aggregations around TSS regions. There was little difference between TSS plots of *sams-1* or *sams-4*-independent genes at either temperature (***Figure 3H1***). However, in basal conditions, Control peaks that depended on *sams-1* had more marked TSS localization (***Figure 3J***), demonstrating that *sams-1* and *sams-4* dependent peaks have distinct TSS architectures. TSS localization was low in all 37 °C samples, following the general trend of decrease after heat shock (***Figure 3K***). We next separated Control peaks into those that were generally SAM synthase-dependent and those that were specific to loss of *sams-1* or *sams-4*. Aggregation of these peaks shows that peaks in Control 15 °C samples that were lost only in *sams-4* RNAi also had the lowest levels of H3K4me3 in TSS regions, whereas promoters that lost this modification only after *sams-1* RNAi had higher levels of H3K4me3 (***Figure 3—figure supplement 2G***). Control 37 °C samples exhibited a similar pattern, with a lower H3K4me3 level overall consistent with what we have observed in heat shock samples (***Figure 3—figure supplement 2H***). Thus, genome wide H3K4me3 contain multiple populations with distinct TSS patterns. Peaks that are present even when *sams-1* or *sams-4* are depleted have the highest levels, whereas *sams-1*-dependent peaks have moderate H3K4me3, and peaks that are lost after *sams-4* RNAi have the lowest levels. Taken together, this shows that individual SAM synthases are linked to distinct sets of H3K4me3 within the genome.

## RNAi of *sams-1* or *sams-4* has similar effects on TSS peaks at tissue-specific genes

Our C&T and RNA seq assays were performed on whole animals. While *sams-1* and *sams-4* are co-expressed in the intestine and hypodermis, which are major stress-responsive tissues, the germline nuclei contain only *sams-4* (***Figure 1B*** and ***Figure 1—figure supplement 1B***). This aligns with our previous observations that *sams-1(RNAi)* animals had normal patterns of H3K4me3 in germline nuclei (***Ding et al., 2015***), whereas RNAi of *sams-4* abrogates H3K4me3 staining in germline nuclei (***Figure 3—figure supplement 3A***). However, embryo production and development appear broadly normal in *sams-4* RNAi embryos (not shown). In order to determine how H3K4me3 might align with tissue-specific expression patterns, we aggregated peaks from tissue-specific RNA seq data published by ***Serizay et al., 2020***. Serizay et al. separated nuclei based on tissue specific GFP expression and defined gene sets that were expressed that were ubiquitously, as well as those that were present only in a single tissue. They also performed ATAC seq (Assay for Transposase-Accessible Chromatin using sequencing). Serizay, et al. defined transcripts by expression pattern and defined sets that were specific to (*tissue*_only), or represented in across multiple tissues (*tissue*_all). ubiquitious_all and Germline_only genes had the most defined patterns of open chromatin around TSSs (***Serizay et al., 2020***). We compared our C&T data with Ubiquitious_all, Germline_only and Intestine_only genes and found

that we identified peaks for around half of these genes in Control RNAi animals at 15 °C or 37 °C (*Figure 3—figure supplement 3B–D*). We found the ubiquitous_all and germline_only genes also had strong H3K4me3 peaks that were reduced equally by *sams-1* or *sams-4* RNAi in both temperature conditions (*Figure 3—figure supplement 3D, G; E, H*). Intestine_only genes showed lower TSS enrichment but were similarly reduced after *sams-1* or *sams-4* RNAi (*Figure 3—figure supplement 3H, I*). These data suggest that differences in germline expression for *sams-1* and *sams-4* are not sufficient to explain differential effects on H3K4me3 peak populations.

## Poor expression of heat shock gene suite in *sams-4(RNAi)* animals

H3K4me3 is found at the promoters of many actively transcribed genes, but it is not necessarily required for gene expression (*Bannister and Kouzarides, 2011*). However, studying chromatin modification in stress responses may reveal additional regulatory effects (*Weiner et al., 2012*). We previously found using ChIP-PCR in the context of the stress response in *C. elegans* that H3K4me3 increased at promoters of genes that responded to bacterial stress in a *sams-1*-dependent manner (*Ding et al., 2015*). However, during the stress response, H3K4me3 did not change at multiple non-stress responsive genes, suggesting that stress-responsive loci might be more sensitive to SAM levels (*Ding et al., 2015*). In order to identify genes that changed in SAM-deficient animals, we performed RNA seq, then compared genes induced by heat shock in control and *sams-1(RNAi)* (*Ding et al., 2018*) with genes induced in *sams-4(RNAi)* animals (*Supplementary file 4*: A-C). Upregulated genes for control and *sams-1(RNAi)* animals appeared closely grouped in principal component analysis, with sams-4(RNAi) upregulated genes and all downregulated gene sets forming distinct groups (*Figure 4—figure supplement 1A*). We previously noted that while *sams-1(RNAi)* animals could not mount the full transcriptional response to bacterial stress, most genes activated by heat increased similarly to controls (*Ding et al., 2018*). *sams-4(RNAi)* animals, in contrast, activate less than 25% of the genes induced by heat in control animals (*Figure 4A*). *sams-1(RNAi)* and *sams-4(RNAi)* animals also induce more that 600 genes in response to heat that are SAM-synthase-specific and which do not increase in control animals (*Figure 4A*). WormCat pathway analysis shows that *sams-4(RNAi)* animals lack the robust enrichment in STRESS RESPONSE (Cat1) and STRESS RESPONSE: Heat (Cat2) evidenced in both Control and *sams-1(RNAi)* samples (*Figure 4B*; *Supplementary file 4*: D-F). In addition, enrichment of the CHAPERONE, PROTEOLYSIS PROTEOSOME categories occurring in *sams-1(RNAi)* animals does not occur after *sams-4(RNAi)*, reflecting lack of induction of these genes which could be important for proteostasis in the heat shock response (*Figure 4C*). Thus, reduction in *sams-1* or *sams-4* results in distinct gene expression programs in both basal conditions (*Figure 1—figure supplement 1A and B*) and during the heat stress response (*Figure 4A–C*). This differentiation of gene expression programs clearly shows that *sams-1* and *sams-4* have distinct functional roles.

Gene expression changes occurring after *sams-1* or *sams-4* depletion could result from direct effects on H3K4me3 or other potential methylation targets, or from indirect effects. Evaluating the impact H3K4me3 on gene expression is also complex, as this modification is generally associated but not necessary for expression of actively transcribed genes *Bannister and Kouzarides, 2011*. In our analysis of H3K4me3 peaks during the heat stress response, we found evidence of multiple peak populations that depend on or occur independently of *sams-1* or *sams-4* (*Figure 3H–K*, *Figure 3—figure supplement 3A-F*). We reasoned, therefore, that it was also critical to determine H3K4me3 levels at *sams-1*- or *sams-4*-dependent genes in the heat shock response.

First, we examined H3K4me3 peak levels at genes with increased in Control RNAi, *sams-1(RNAi)* or *sams-4(RNAi)* during heat shock. We found that genes dependent on *sams-1* or *sams-4* in the heat shock response were marked by lower overall H3K4me3 levels at the TSSs (*Figure 4D*). However, this analysis included large numbers of upregulated genes in *sams-1* or *sams-4* outside of the wildtype heat stress response. Therefore, we next focused on genes normally upregulated during heat shock and divided them according to SAM synthase dependence. Strikingly, isolating the *sams-1*-dependent genes revealed a strong peak 5′ to the TSS, which was not evident in the larger subset of Control or *sams-4(RNAi)*-dependent upregulated genes (*Figure 4E and F*). Among the genes with robust peaks in heat-shocked *sams-1(RNAi)* animals were two F-box proteins, *fbxa-59* and T27F6.8, which were robustly expressed in *sams-1* but not *sams-4* animals (*Figure 4C–*). Downregulation of T27F6.8 did not affect the survival of the animals after heat shock (*Figure 4—figure supplement 1B*) while survival of animals fed *fbxa-59* RNAi was modestly affected (*Figure 4—figure supplement 1C*). Survival in heat

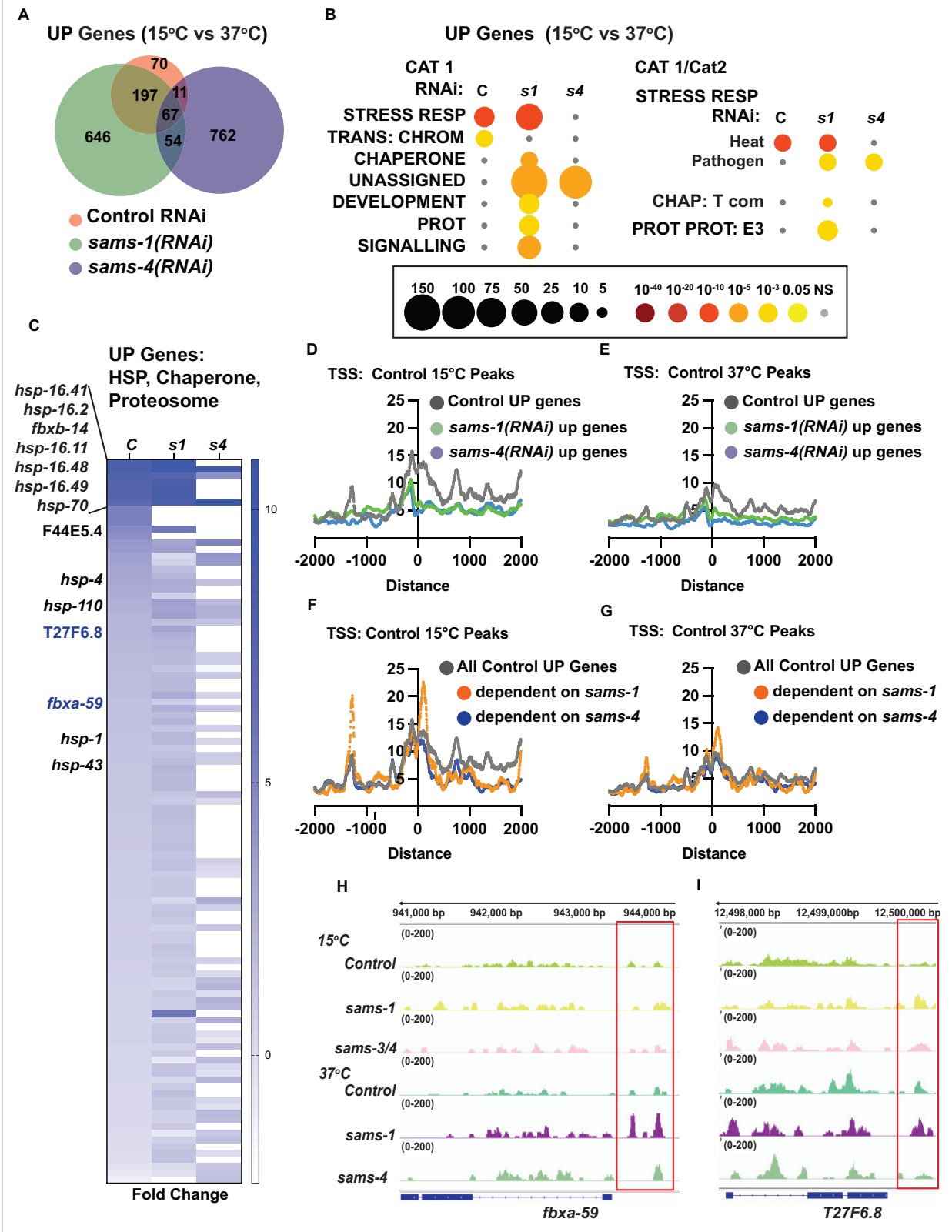

**Figure 4.** Distinct gene expression and H3K4me3 patterns after heat shock in *sams-1* and *sams-4* RNAi animals. (**A**) Venn diagram showing overlap of genes upregulated by heat shock in control, *sams-1* or *sams-4* RNAi animals. *sams-1* data is from ***Ding et al., 2018***. (**B**) Bubble charts show broad category enrichment of up genes determined by Worm-Cat in control (RNAi) or *sams-1* or *sams-4* animals in genes changed (FDR <0.01) after heat shock. (**C**) Heat map for heat shock response genes upregulated following heat shock in animals fed control RNAi, *sams-1* or *sams-4(RNAi)*. TSS plots

*Figure 4 continued on next page*

*Figure 4 continued*

showing aggregation of H3K4me3 in genes upregulated in control, *sams-1* or *sams-4* RNAi at (**D**) 15 °C or (**E**) 37 °C. TSS plots showing aggregation of H3K4me3 in all genes upregulated in control or *sams-1* dependent or *sams-4* RNAi dependent at (**F**) 15 °C or (**G**) 37 °C. The Y axis on TSS plots shows Peaks per base pair of gene. Genome browser tracks for (**H**) *fbxa-59* and (**I**) *T27F6.8* to visualize changes in H3K4me3 enrichment in animals fed control, *sams-1* or *sams-4(RNAi)* at 15 °C or 37 °C.

The online version of this article includes the following figure supplement(s) for figure 4:

**Figure supplement 1.** *sams-1* and *sams-4* have distinct gene expression patterns after heat shock.

shock may be multi-genic and rely on pathway responses rather than single genes. However, our data reveals genes upregulated in the heat shock response may have different H3K4me3 levels depending on requirements for *sams-1* or *sams-4*. In addition, our results suggest that roles for H3K4me3 may become clearer when genome-wide methylation populations are separated into biologically responsive categories.

## SAM synthase-specific effects on genes downregulated in the heat shock response

Transcriptional responses to heat shock largely focus on rapidly induced genes that provide protection from changes in proteostasis (*Morimoto, 2006*; *Mahat et al., 2016*). However, downregulated genes

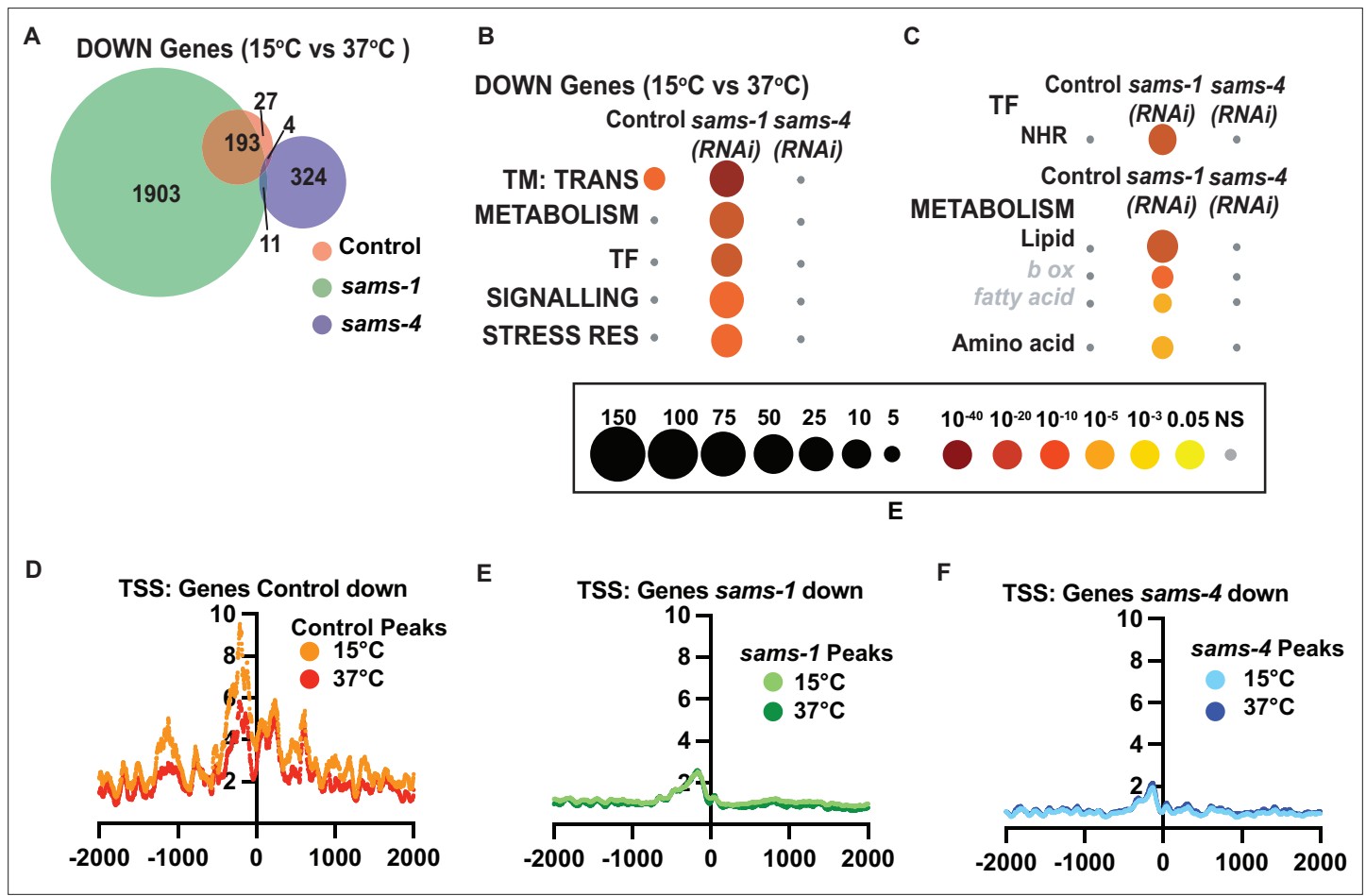

**Figure 5.** Genes that depend on *sams-1* or *sams-4* for expression have reduced H3K4me3. (**A**) Venn diagram showing overlap in downregulated genes in animals fed control, *sams-1*, or *sams-4(RNAi)* at 37 °C. (**B**) Bubble charts show broad category enrichment of metabolism genes determined by Worm-Cat in *sams-1* or *sams-4* animals in genes changed (FDR <0.01) after heat shock. (**C**) Bubble charts show broad category enrichment of transcription factor and metabolism genes determined by Worm-Cat in *sams-1* or *sams-4* animals in genes changed (FDR <0.01) after heat shock. Aggregation plots showing average enrichment of reads around the transcription start site (TSS) in animals fed (**D**) control, (**E**) *sams-1* or (**F**) *sams-4(RNAi)* at 15 °C or 37 °C. The Y axis on TSS plots shows Peaks per base pair of gene.

could also play important roles. For example, the WormCat category of TRANSMEMBRANE TRANS-PORT (TM TRANS) is enriched in genes downregulated after heat shock in *C. elegans* (*Figure 5A and B*). Previously we observed that heat shocked animals depended on *sams-1* for normal expression of nearly 2000 genes, falling within WormCat Categories of METABOLISM, TRANSCRIPTION FACTOR (TF), SIGNALLING, and STRESS RESPONSE (*Ding et al., 2018*; *Figure 5A and B*). Interestingly, the metabolic genes dependent on *sams-1* include those in lipid metabolism, whereas the TF enrichment was centered around nuclear hormone receptors (NHRs) (*Figure 5C and D*), which regulate many metabolic and stress responsive genes in *C. elegans* (*Arda et al., 2010*). However, neither the shared TM TRANSPORT nor the *sams-1* specific categories depend on *sams-4* (*Figure 5B and C*). Thus, as in genes upregulated during the heat shock response, genes downregulated in the heat shock response also have differential requirements for *sams-1* and *sams-4*.

Next, we examined H3K4me3 levels around TSSs of genes that lost expression during heat shock in Control, *sams-1* or *sams-4(RNAi)* animals. Genes decreasing in Control animals had a slight reduction of H3K4me3 peaks when comparing 15°C and 37 °C samples, consistent with global levels after heat shock (*Figure 5D*). RNAi of *sams-1 or sams-4* also broadly reduced H3K4me3 TSS enrichment at downregulated genes (*Figure 5D–F*). However, there were minimal differences before or after heat shock, suggesting expression patterns affecting survival could be established before induction of the stress.

H3K4me3 has been reported to act as a bookmarking modification, therefore we hypothesized that some loci could be affected before heat shock, with expression changing afterward. Therefore, we more closely examined genes with *sams-1*-dependent H3K4me3 at 15 °C that lost expression during heat shock. Those genes were highly enriched for METABOLISM: Lipid: beta oxidation and NHR transcription factors (*Figure 6A*). We noted they included multiple members of a regulatory circuit that control expression of a beta-oxidation-like pathway that degrades toxic fatty acids identified by the Walhout lab (*Bulcha et al., 2019*), including *nhr-68, nhr-114* and beta-oxidation genes *acdh-1, hach-1 ech-6,–8,* and *–9* (*Figure 6B and C*). Indeed, *nhr-68*, the initiating TF in this regulatory circuit, shows lower levels of H3K4me3 at its promoter in basal conditions, compared to Control or *sams-4* RNAi animals (*Figure 6D*). The H3K4me3 peak overlaps with another gene, *pms-2*, whose expression does not change after heat shock or upon SAM synthase RNAi (*Supplementary file 4*: Tabs A-C). In order to test if H3K4me3-dependent regulation of *nhr-68* was important for survival during heat shock, we made use of a construct expressing *nhr-68* under the intestine-specific *ges-1* promoter (*Bulcha et al., 2019*), where H3K4me3 peaks do not change after RNAi of *sams-1* or *sams-4* (*Figure 6—figure supplement 1A*). Expression of *nhr-68* under this heterologous promoter had a moderate, but significant effect on survival (*Figure 6E*). Thus, downregulation of *nhr-68* in sams-1 animals after heat shock could be part of a program enhancing survival (*Figure 6—figure supplement 1B*). Taken together, our results suggest differences in H3K4me3 patterns in *sams-1* and *sams-4* animals before heat shock may also influence gene expression patterns during the stress response. This demonstrates that *sams-1* and *sams-4* are required for distinct sets of genes in the heat stress response and contribute to different H3K4me3 patterns.

## Discussion

The molecules that modify chromatin are produced by metabolic pathways (*Cheng and Kurdistani, 2022*). Use of ATP, AcetylCoA or SAM for phosphorylation, acetylation or methylation of histones is tightly regulated and many studies have focused on control of enzymes or enzyme-containing complexes. Acetylation and methylation may also be regulated by metabolite levels (*Hsieh et al., 2022*; *Wellen et al., 2009*). This allows the chromatin environment to sense and respond to changes in key metabolic pathways. However, effects of methylation on chromatin are multifaceted: DNA and H3K9me9 have strong repressive effects, whereas other modifications such as H3K4me3 and H3K36me3 are associated with active transcription (*Bannister and Kouzarides, 2011*). These marks, especially H3K4me3, are most sensitive to SAM levels, most likely due to the kinetics of the H3K4me3 MTs (*Mentch and Locasale, 2016*). SAM is an abundant metabolite that contributes to multiple biosynthetic pathways in addition to acting as the major donor for histone, DNA and RNA methylation (*Walsh et al., 2018*). Reduction in SAM levels has major phenotypic consequences in animals, altering lipid levels in murine liver and *C. elegans* (*Walker et al., 2011*; *Lu et al., 2001*), altering differentiation potential in iPS cells (*Shyh-Chang et al., 2013*) and changing stress resistance (*Ding et al., 2018*). In

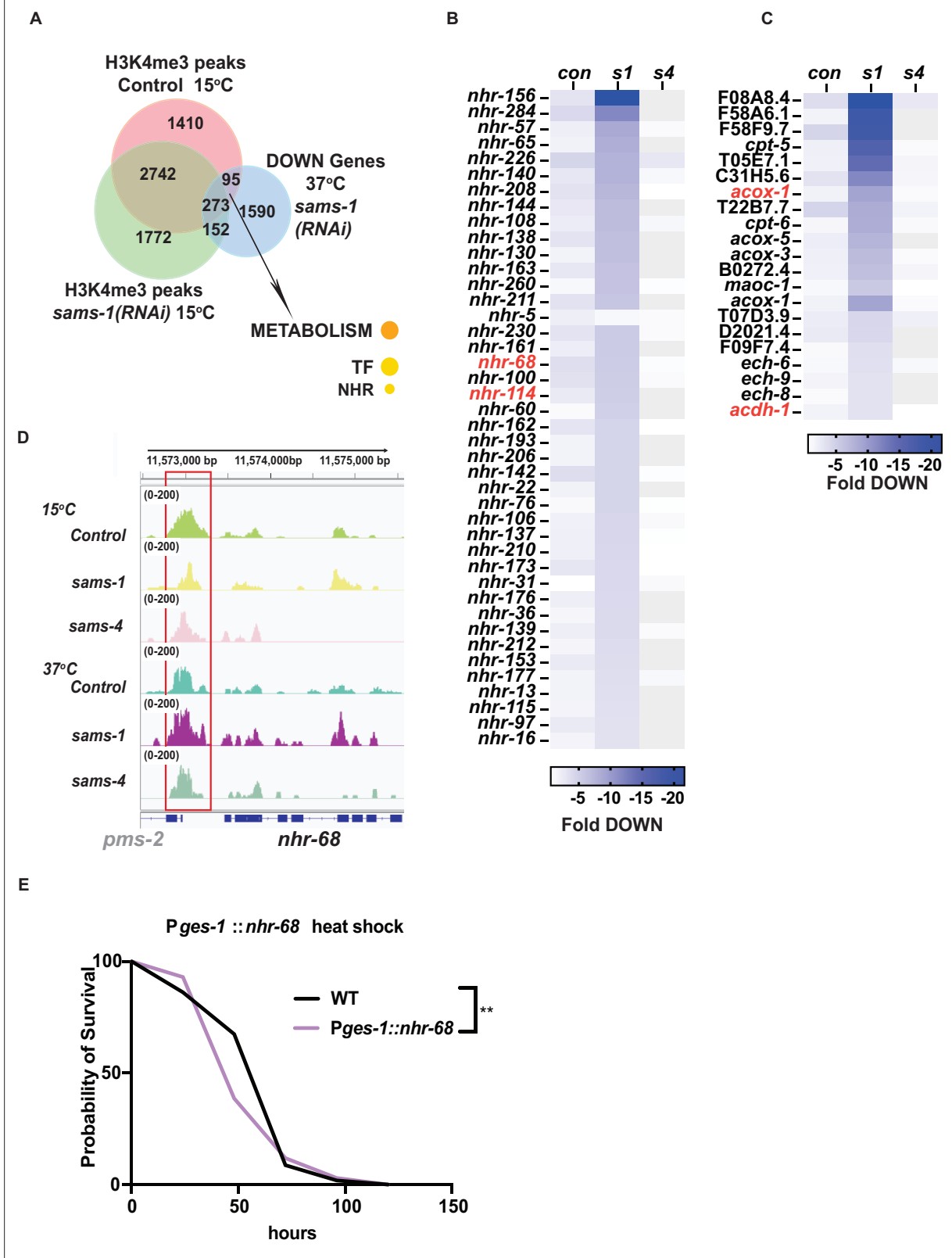

**Figure 6.** *nhr* and lipid beta oxidation genes lose H3K4me3 after *sams-1* RNAi but expression after heat shock. (**A**) Venn diagram showing the overlap between H3K4me3 peaks identified in animals fed control or *sams-1(RNAi)* at 15 °C and downregulated genes identified in heat shocked animals fed *sams-1(RNAi)*. Heat map for (**B**) nuclear hormone response genes and (**C**) lipid β-oxidation genes downregulated following heat shock in animals fed

*Figure 6 continued on next page*

*Figure 6 continued*

control RNAi, *sams-1* or *sams-4(RNAi)*. Genes linked to nhr-68 feedback loop (**Bulcha et al., 2019**) are marked in red. (**D**) Genome browser tracks for *nhr-68* to visualize changes in H3K4me3 enrichment in animals fed control, *sams-1,* or *sams-4(RNAi)* at 15 °C or 37 °C.

The online version of this article includes the following figure supplement(s) for figure 6:

**Figure supplement 1.** Schematic of potential *nhr-68* module regulation in sams-1 animals.

addition, 1 CC has been identified as a causal regulator of aging **Annibal et al., 2021** and is important in cancer development (**Sullivan et al., 2021**; **Gao et al., 2019**). However, the abundance of SAM and its targets have made it difficult to connect changes in methylation to molecular pathways regulating these physiological effects. In addition, studying effects of SAM is difficult in culture because SAM itself is labile (**Sun and Locasale, 2022**) and tissue culture media is replete with 1 CC metabolites (**Sullivan et al., 2021**). Important insights have been made into the impact of SAM on the breadth of H3K4me3 peaks using methionine depletion (**Mentch et al., 2015**; **Tang et al., 2017**; **Dai et al., 2018**); however, this approach could affect other pathways. In this study, we have taken the approach of limiting SAM synthase expression in *C. elegans*, then using genetic and molecular approaches to link methylation-dependent pathways to changes in stress responses. We found that individual SAM synthases could have distinct effects even on a single methylation target such as H3K4me3. This observation not only shows that examining how SAM is produced within the cells allows differentiation of phenotypic effects, but also supports the striking notion of 'where' SAM comes from affects its functional output. While mammalian cells express either one of two SAM synthases, MAT2A, which is present in non-liver cells, may be present in multiple regulatory isoforms (**Murray et al., 2019**). Thus, the isoform-specific production and functional targets for SAM synthases we uncover could also be important in mammals. Hints of this exist in other cellular systems – 1 CC enzymes, for example, have been associated with chromatin modifying complex in yeast (**Li et al., 2015**) and mammalian cells (**Greco et al., 2020**).

H3K4me3 is clearly an important link between SAM levels, aging and stress phenotypes, as loss or reduction of H3K4 MT function phenocopy aspects of SAM depletion (**Ding et al., 2015**; **Ding et al., 2018**). However, this modification is also wide-spread, and transcription may occur even when this mark is not present (**Hödl and Basler, 2012**). By studying acute changes in gene expression during heat stress response in *C. elegans*, we have found that H3K4me3 populations can also be separated based on SAM synthase requirements. The importance of H3K4me3 during heat shock is also reflected in the interactions between the SAM synthases and the KMTs/KDMTs as lowering levels of *set-2* or *set-16* increase survival. This suggests that the context of low SAM from SAMS-1, reducing H3K4me3 can have additional benefits. Future studies identifying genomic targets of H3K4me3 KMTs together with SAM synthases may be important for untangling these effects.

SAM synthase-specific effects may also vary according to the biological context, as loss of *sams-1* improves the ability of *C. elegans* to survive heat stress, while limiting its ability to withstand bacterial pathogens (**Ding et al., 2018**). Our previous studies showed that the induction of bacterial pathogen induced genes was limited in the absence of *sams-1*, however, in this study, we find links between *sams-1*-dependent genes in basal conditions and effects on survival after heat shock. Thus, the altered methylation landscape in *sams-1* animals provides a context favorable to extended lifespan and survival in heat stress but which limits other stress responsive genes. This context may depend on systems level effects and not on a single 'target' gene, as our analysis of genes that lose peaks in *sams-1* or *sams-4(RNAi)* animals have modest effects, but do not recapitulate the entire phenotype. It is also possible that there are genes or specific modules that drive enhanced survival in *sams-1* animal or responsible for viability after loss of *sams-4(RNAi)*. Our approach dividing peaks into groups based on responsiveness to *sams-1* or *sams-4* demonstrates the importance of identifying specific populations of H3K4me3; combining *set-2* or *set-16* sensitive loci may provide the resolution to identify these loci in future studies. Manipulation of the 1 CC is of interest as a modulator of aging (**Annibal et al., 2021**) and affects multiple biological processes. Our studies demonstrate that lowering SAM, or reducing levels of a key methylation target such as H3K4me3, does not represent a single biological state and that it is important to consider that effects may depend on synthase-specific regulation or context. Future identification of these regulators will provide the mechanistic details key to understanding the role of the 1 CC in aging and stress.

## Limitations

The genetic tools used in our study provide a method to reduce SAM from a specific enzymatic source. However, the roles for SAM in the cell are broad and can affect methylation of multiple targets. While our metabolomics assays show that SAM increases in heat shocked *sams-1(RNAi)* animals, we have not demonstrated that this SAM is derived from *sams-4*. In addition, survival benefits after heat shock occur across broad cellular functions including proteostasis and other methylation marks such as H3K9me3 (*Das et al., 2021*). Thus, there may be multiple additional methylation-dependent mechanisms that influence survival of *sams-1* or *sams-4* animals during heat shock. In addition, we measured gene expression and H3K4me3 at 2 hr post heat shock, whereas the survival assay occurs over multiple days. Thus, there may be changes in gene expression or histone modifications occurring at later times that also affect survival.

# Materials and methods

**Key resources table**

| Reagent type (species) or resource | Designation | Source or reference | Identifiers | Additional information |
|---|---|---|---|---|
| Gene (*C. elegans*) | *sams-1* | Wormbase | WBGene00008205 | |
| Gene (*C. elegans*) | *sams-4* | Wormbase | WBRNAi00010322 | |
| Gene (*C. elegans*) | *set-2* | Wormbase | WBGene00004782 | |
| Gene (*C. elegans*) | *set-16* | Wormbase | WBGene00011729 | |
| Gene (*C. elegans*) | *rbr-2* | Wormbase | WBGene00004319 | |
| Gene (*C. elegans*) | *spr-5* | Wormbase | WBRNAi00004611 | |
| Gene (*C. elegans*) | *lsd-1* | Wormbase | WBGene00011615 | |
| Genetic reagent (*C. elegans*) | N2 | *Caenorhabditis* Genetics Centre | Wild type | |
| Genetic reagent (*C. elegans*) | *sams-1(ok3033)* | CGC | HA1975 | sams-1(ok3033) X |
| Genetic reagent (*C. elegans*) | *sams-4(ok3315)* | CGC | RB2420 | C06E7.3(ok3315) IV |
| Genetic reagent (*C. elegans*) | *rfp::sams-1(ker5); sams-1(ok3033)* | This paper | WAL500 | WAL500. See 'Materials and methods, section *C. elegans* strains' |
| Genetic reagent (*C. elegans*) | *gfp::sams-4(ker6); sams-4(ok3315)* | This paper | WAL501 | WAL501; See 'Materials and methods, section *C. elegans* strains' |
| Genetic reagent (*C. elegans*) | Pges-1::NHR-68::GFP | Bulcha, J.T Cell Reports 26, 460–468.e4. 10.1016 /j.celrep.2018.12.064. | VL1296 | See 'Materials and methods, section *C. elegans* strains' |
| Genetic reagent (*C. elegans*) | *sams-3::mKate(nu3139)* | This paper, InVivo Biosystems | WAL503 | WAL503; See 'Materials and methods, section *C. elegans* strains' |
| Genetic reagent (*C. elegans*) | *rfp::sams-1;gfp(ker5)::sams-4(ker6)* | This paper | WAL502 | See 'Materials and methods, section *C. elegans* strains' |
| Genetic reagent (*C. elegans*) | *Escherichia coli OP50* | CGC | N/A | Used as food source for *C. elegans* |
| Strain, strain background (*E. coli*) | *sams-1* RNAi | Source Bioscience | X-5P21 | Used to knock down target mRNA |
| Strain, strain background (*E. coli*) | *sams-4* RNAi | Source Bioscience | IV-3C01 | Used to knock down target mRNA |
| Strain, strain background (*E. coli*) | *set-2* RNAi | ORFeome RNAi library | mv_C26E6.9 | Used to knock down target mRNA |

*Continued on next page*

*Continued*

| Reagent type (species) or resource | Designation | Source or reference | Identifiers | Additional information |
|---|---|---|---|---|
| Strain, strain background (*E. coli*) | *set-16* RNAi | Source Bioscience | III-6D12 | Used to knock down target mRNA |
| Strain, strain background (*E. coli*) | *rbr-2* RNAi | Source Bioscience | IV-5D22 | Used to knock down target mRNA |
| Strain, strain background (*E. coli*) | *spr-5* RNAi | Source Bioscience | I-6H02 | Used to knock down target mRNA |
| Strain, strain background (*E. coli*) | *lsd-1 RNAi* | Source Bioscience | X-5P17 | Used to knock down target mRNA |
| Antibody | Tri-Methyl-Histone H3 Lys4 (Rabbit monoclonal) | Cell Signaling Technology | C42D8 | Used for IF (1:200) and CUT&Tag (1:50) |
| Software, algorithm | GraphPad Prism v8 | https://www.graphpad.com | N/A | Used for statistical analysis of data and generate graphs |
| Software, algorithm | HOMER | https://homer.ucsd.edu/homer/, *Heinz et al., 2010* | | Bioinformatic data analysis software |
| Software, algorithm | Dolphin | https://www.umassmed.edu/biocore/introducing-dolphin/, *Yukselen et al., 2020* | | Bioinformatic data analysis software |
| Software, algorithm | DeBrowser | https://debrowser.umassmed.edu, *Kucukural et al., 2019* | | Bioinformatic data analysis software |
| Software, algorithm | ChipPeakAnno | https://bioconductor.org/packages/release/bioc/html/ChIPpeakAnno.html, *Zhu et al., 2010* | | Bioinformatic data analysis software |
| Software, algorithm | BioVenn | https://biovenn.nl, *Hulsen et al., 2008* | | Used to generate Venn diagrams |
| Software, algorithm | WormCat | https://www.wormcat.com/, *Higgins et al., 2022* | | Used to generate WormCat data |
| Software, algorithm | BowTie2 | https://bowtie-bio.sourceforge.net/bowtie2/index.shtml, *Langmead et al., 2009* | | Bioinformatic data analysis software |
| Commercial assay or kit | KAPA Library Quantification Kits | https://sequencing.roche.com/global/en/products/group/kapa-library-quantification-kits.html#productInfo | N/A | Used to quantify DNA for CUT&Tag |
| Commercial assay or kit | NEBNext High-Fidelity 2 X PCR Master Mix | https://www.neb.com/products/m0541-nebnext-high-fidelity-2x-pcr-master-mix#Product%20Information | N/A | Used to amplify libraries in CUT&Tag |
| Chemical compound, drug | ConA beads | https://www.bangslabs.com/sites/default/files/imce/docs/PDS%20720%20Web.pdf | N/A | Used in CUT&Tag 'See Materials and methods section; CUT&Tag' |
| Other | Potter-Elvehjem Tissue Grinder With PTFE Pestle | https://www.thomassci.com/Equipment/Grinders/_/POTTER-ELVEHJEM-TISSUE-GRINDER-WITH-PTFE-PESTLE?q=Kontes%20Glass | 3432 N59; Mfr. No. 886000–0020 | Used for homogenizing worms in CUT&Tag 'Materials and methods section CUT&Tag' |

### *C. elegans* strains

N2(*Caenorhabditis* Genetics Center); *sams-1(lof)(ok3033); sams-3(ok2932) IV*, *sams-4((ok3315)* IV, *Caenorhabditis* Genetics Center), tagRFP::SAMS-1 (WAL500, this study); GFP::SAM-4(WALK501, this study); SAMS-1::RFP;GFP::SAMS-4(WAL502, this study), SAMS-3::mKate (WAL305). *Pges-1*::NHR-68::GFP (VL1296) (*Bulcha et al., 2019*). CRISPR tagging for WAL500 and WAL501 were done by the UMASS Medical School transgenic core, confirmed by PCR for genotype and outcrossed three times to wild type animals. Next, each strain was crossed to the respective deletion allele to create WAL503 (RFP::*sams-1(ker5); sams-1(ok3033)*) and WAL504(GFP::*sams-4(ker6); sams-4(ok3315)*). *sams-3::mKate(nu3139)* (COP2476) was constructed using CRISPR by In Vivo biosystems then outcrossed three times (WAL305).

### *C. elegans* culture, RNAi and stress applications

*C. elegans* (N2) were cultured using standard laboratory conditions on *E. coli* OP50 or HT115 expressing appropriate RNAi. RNAi expression was induced using 6 mM IPTG. Adults were bleached onto RNAi plates and allowed to develop to the L4 to young adult transition before stresses were applied. For heat stress applications, animals were raised at 15 °C from hatching then at the L4/young adult transition replicate plates were placed at 15 °C or 37 °C for 2 hr. After each stress, animals were washed off the plates with S-basal, then pellets frozen at –80 °C. RNA was prepared as in *Ding et al., 2015*. For survival assays, ~10–15 adult N2 animals were bleached on 60 mm RNAi plates. The eggs were allowed to hatch and grow to young adults at 15 °C. Twenty-five to 30 young adults were then moved to 35 mm plates in triplicate (75–90 animals per RNAi treatment) and subjected to heat shock at 37 °C for 2 hr. Animals were kept at 20 °C for the remainder of the assay. Dead animals were identified by gentle prodding, were counted and removed each day. Animals that died of bagging or from desiccation on the side of the plate were not counted. Three independent non blinded biological replicates were carried out and Kaplan-Meir curves were generated with GraphPad Prism v8.0. For lifespan experiments, the N2 adults were bleached on 60 mm RNAi plates. The eggs were allowed to hatch and grow to young adults at 20 °C. Twenty-five to 30 young adults were then moved to 35 mm plates in triplicate (75–90 animals per RNAi treatment). Adults were moved to fresh plates every day and dead animals were identified by gentle prodding and removed each day. Three independent non blinded biological replicates were carried out and Kaplan-Meir curves were generated with GraphPad Prism v8.0.

### Gene expression analysis, RNA sequencing and analysis

RNA for deep sequencing was purified by Qiagen RNAeasy. Duplicate samples were sent for library construction and sequencing at BGI (China). Raw sequencing reads were processed using an in-house RNA-Seq data processing software Dolphin at University of Massachusetts Medical School (*Yukselen et al., 2020*). The raw read pairs were first aligned to *C. elegans* reference genome with ws245 annotation. The RSEM method was used to quantify the expression levels of genes and Deseq was used to produce differentially expressed gene sets with more than a twofold difference in gene expression, with replicates being within 0.05 in a Students T test and a False Discovery Rate (FDR) under 0.01. Statistics were calculated with DeBrowser (*Kucukural et al., 2019*). Venn Diagrams were constructed by BioVenn (*Hulsen et al., 2008*). WormCat analysis was performed using the website https://www.wormcat.com/ (*Holdorf et al., 2020*; *Higgins et al., 2022*) and the whole genome annotation version 2 (v2) and indicated gene sets. PCA was conducted by using *prcomp* in R and graphed with *ggplot* in R studio.

### Immunofluorescence

For H3K4me3 (Cell Signaling Technology, catalogue number C42D8) staining, dissected intestines were incubated in 2% paraformaldehyde, freeze cracked, then treated with –20°C ethanol before washing in PBS, 1% Tween-20, and 0.1% BSA. Images were taken on a Leica SPE II at identical gain settings within experimental sets. Quantitation was derived for pixel intensity over nuclear area for at least seven dissected intestines, with at least three nuclei per intestine. Three biological repeats were carried out for every experiment.

### Sample preparation for LC-MS

*C. elegans* (N2) gravid adults (~15–20) were bleached onto 60mm RNAi plates, eggs were allowed to hatch and grow to young adults at 15°C. For heat stress application, replicate plates were placed at either 15°C or 37°C for 2 hr. At the end of the heat stress, worms were collected in S-Basal, and pellets were frozen at –80°C. Four independent biological replicates were collected. To prepare the samples for LC-MS, the pellet was thawed on ice and washed with 0.9% NaCl. Washed worms were then transferred to 2 mL FASTPREP tubes (MP Biomedicals) containing 1.4 mm ceramic beads (Qiagen). The samples were then resuspended in 1 mL 80% methanol (LC-MS grade) and homogenized using a bead beater (6.5 m/s; 20 s). The samples were cooled on ice between cycles. The homogenized samples were then vortexed at 4 °C for 10 min and centrifuged at 21,000 RPM for 10 min at 4 °C. The supernatant was removed at dried under vacuum. The pellet was resuspended in ice cold RIPA buffer and vortexed at 4 °C for 10 min and centrifuged at 21,000 RPM at 4 °C for 10 min. The supernatant

was removed and used for protein quantification using Pierce Protein BCA assay kit (ThermoFisher). The protein quantification was then used to resuspend the pellet for an equal input of 0.5 µg/ml of protein per sample.

## LC-MS analysis

### Absolute quantification of SAM

Samples were extracted in 80% methanol containing 500 nM methionine-$^{13}C_5$-$^{15}N$ (Cambridge Isotope Laboratories, Inc) as an internal standard and metabolites were detected as described above. Absolute quantification of SAM was performed using an external calibration curve prepared with synthetic standard, and peak areas were normalized to methionine-$^{13}C_5$-$^{15}N$. Normalized peak areas from the standard curve were fit to a quadratic log-log equation with an $r^2$ value of >0.995 which was then used to calculate the concentration of SAM in each sample. Statistical analysis was carried out for the data using GraphPad Prism (v8.0).

## Relative metabolite profiling

Metabolomics was conducted on a QExactive Plus bench top orbitrap mass spectrometer equipped with an Ion Max source and a HESI II probe, which was coupled to a Vanquish Horizon HPLC system (Thermo Fisher Scientific, San Jose, CA). External mass calibration was performed using the standard calibration mixture every 7 days. Dried extracts were reconstituted in enough water to achieve a final concentration of 0.5 µg/ml protein per sample. Two µL of this resuspended sample were injected onto a SeQuant ZIC-pHILIC 150x2.1 mm analytical column equipped with a 2.1x20 mm guard column (both 5 mm particle size; Millipore Sigma). Buffer A was 20 mM ammonium carbonate, 0.1% ammonium hydroxide; Buffer B was acetonitrile. The autosampler tray was held at 4°C. The chromatographic gradient was run at a flow rate of 0.150 mL/min as follows: 0–20 min: linear gradient from 80% to 20% B; 20–20.5 min: linear gradient form 20% to 80% B; 20.5–28 min: hold at 80% B. The mass spectrometer was operated in full-scan, polarity-switching mode, with the spray voltage set to 4.0 kV, the heated capillary held at 320°C, and the HESI probe held at 350°C. The sheath gas flow was set to 10 units, the auxiliary gas flow was set to 1 units, and the sweep gas flow was set to 1 unit. MS data acquisition was performed in a range of $m/z$=70–1000,, with the resolution set at 70,000, the AGC target at $1x10^6$, and the maximum injection time at 20 ms. An additional scan ($m/z$ 220–700) in negative mode only was included to enhance detection of nucleotides. Relative quantitation of polar metabolites was performed TraceFinder 5.1 (Thermo Fisher Scientific) using a 5 ppm mass tolerance and referencing an in-house library of chemical standards. Statistical analysis was carried out for the data using GraphPad Prism (v8.0).

## CUT&Tag

*C. elegans* (N2) were cultured using standard laboratory conditions on *E. coli* OP50. Adults were bleached onto RNAi plates and allowed to develop to the L4 to young adult transition before heat stress was applied. For heat stress applications, animals were raised at 15 °C from hatching then at the L4/young adult transition replicate plates were placed at 15 °C or 37 °C for 2 hr. At the end of the heat stress, animals were washed off the plates with S-basal, then pellets frozen at –80 °C. Worm pellets were washed with S-Basal to remove bacteria, then resuspended in 750 µL of chilled Nuclei Purification Buffer (50 mM HEPES pH = 7.5, 40 mM NaCl, 90 mM KCl, 2 mM EDTA, 0.5 mM EGTA, 0.2 mM DTT, 0.5 mM PMSF, 0.5 mM spermidine, 0.1% tween 20, and cOmplete proteinase inhibitor cocktail (Roche)). The suspension was then transferred to Potter-Elvehjem Tissue Grinder (3 mL). The worms were ground with 2–3 cycles consisting of ~45–50 strokes of the grinder. The samples were chilled on ice for ~5 min between consecutive cycles. The lysates were passed through 100 micron filter (X3) followed by 40 micron (X3) (Pluriselect). The lysates were then centrifuged at 4500 RPM for 10 min at 4 °C. The pellets were resuspended gently in wash buffer (1 M HEPES pH 7.5, 5 M NaCl, 2 M spermidine). Concanavalin bead slurry (10 µL/sample) was added gently to the samples and allowed to incubate at room temperature for 15 min in an end-over-end rotator. The sample tubes were then transferred to a magnetic stand and liquid was gently removed. The nuclei were gently resuspended in 50 µL of chilled antibody buffer (8 µL 0.5 M EDTA, 6.7 µL 30% BSA in 2 mL Dig-wash buffer (400 µL 5% digitonin with 40 mL Wash buffer)). 1 uL anti-H3K4me3 antibody (Cell Signaling Technology, catalogue number C42D8) was added to the suspension and allowed to bind overnight

at 4 °C on a nutator shaker. Samples without any antibody added were used as controls to correct for background reads and further processed per the CUT&Tag protocol *Kaya-Okur et al., 2019* to generate sequencing libraries. The libraries were amplified by mixing 21 μL of DNA with 2 μL each of (10 μM) barcoded i5 and i7 primers, using a different combination for each sample. 25 μL NEBNext HiFi 2×PCR Master mix (NEB) was added, and PCR was performed using the following cycling conditions: 72 °C for 5 min (gap filling); 98 °C for 30 s; 17 cycles of 98 °C for 10 s and 63 °C for 30 s; final extension at 72 °C for 1 min and hold at 4 °C. 1.1×volume of Ampure XP beads (Beckman Coulter) was incubated with libraries for 10 min at room temperature to clean up the PCR mix. Bead bound DNA was purified by washing twice with 80% ethanol and eluting in 20 μL 10 mM Tris pH 8.0. Size distribution of the libraries was determined by Fragment analyzer and concentration by the KAPA Library Quantification Kit before sequencing to determine the H3K4me3 landscape in basal and heat stress condition in worms fed on control, *sams-1* or *sams-4* RNAi. Sequencing of the prepared libraries was carried out on Illumina NextSeq 500.

## Data analysis

Paired end reads from each sample were aligned to the *C. elegans* genome (ce11 with ws245 annotations) using Bowtie2 (*Langmead et al., 2009*) with the parameters -N 1 and -X 2000. Duplicate reads were removed using Picard (RRID:SCR_006525) and the reads with low quality scores (MAPQ <10) were removed. HOMER software suite was used to process the remaining mapped reads (*Heinz et al., 2010*). The 'makeUCSCfile' command was used for generating genome browser tracks. Data was normalized to library size. the 'findPeaks <tag directory> -style histone -o auto' command was used for calling H3K4me3 peaks and the 'annotatePeaks' command was used for making aggregation plots. Differential peak calling was accomplished using (*Zhu et al., 2010*) the command ". We used the findOverlapsOfPeaks command in ChipSeqAnno[37] with a max gap of 1000 basepairs to determine peak overlap. TSS plots were generated using HOMER (*Heinz et al., 2010*) and Venn Diagrams were constructed by BioVenn (*Hulsen et al., 2008*).

Correlation matrices were generated with deeptools version 3.5.1 (*Ramírez et al., 2016*). Multibamsummary was used to compare bam files from each sample, using default values except `--binSize` 2000. This data was visualized using plotCorrelation with --removeOutliers and the Pearson method. Previously published datasets were used to compare H3K4me3 Cut and Tag versus previously published data sets. Young adults fed a normal diet were used from *Wan et al., 2022*. Day 2 *glp-1* adults were chosen from *Pu et al., 2015*. modENCODE ChIP-seq data drew from L3 animals (*Ho et al., 2014*).

## Acknowledgements

We would like to acknowledge the Walker lab for reading of the manuscript, Drs. Marian Walhout and Craig Peterson for helpful discussions and Dr. Marie Bao at Life Science Editors for manuscript assistance. Absolute quantification of SAM was carried out at the Whitehead Metabolomics Core (Cambridge, MA). We thank the UMASS Transgenic animal core (Dr. Paola Perrat and Dr. Michael Francis) for construction of RFP::SAMS-1and GFP::SAMS-4. Funding is from the NIH: 1R01AG053355 to AKW, R01HD072122 to TGF and K99CA273420 to SG.

## Additional information

### Funding

| Funder | Grant reference number | Author |
| --- | --- | --- |
| National Institutes of Health | 1R01AG053355 | Amy K Walker |
| National Institutes of Health | R01HD072122 | Thomas G Fazzio |
| National Institutes of Health | K99CA273420 | Sneha Gopalan |

| Funder | Grant reference number | Author |
| --- | --- | --- |

The funders had no role in study design, data collection and interpretation, or the decision to submit the work for publication.

## Author contributions

Adwait A Godbole, Conceptualization, Data curation, Formal analysis, Validation, Investigation, Visualization, Writing - original draft, Writing - review and editing; Sneha Gopalan, Conceptualization, Formal analysis, Investigation, Writing - review and editing; Thien-Kim Nguyen, Paula Vo, Investigation; Alexander L Munden, Data curation, Software, Formal analysis, Methodology; Dominique S Lui, Investigation, Visualization; Matthew J Fanelli, Data curation, Investigation; Caroline A Lewis, Jessica B Spinelli, Formal analysis, Investigation; Thomas G Fazzio, Conceptualization, Investigation, Writing - review and editing; Amy K Walker, Conceptualization, Data curation, Software, Formal analysis, Supervision, Funding acquisition, Validation, Investigation, Visualization, Methodology, Writing - original draft, Writing - review and editing

## Author ORCIDs

Thomas G Fazzio http://orcid.org/0000-0002-0353-7466
Amy K Walker http://orcid.org/0000-0003-1899-8916

## Decision letter and Author response

Decision letter https://doi.org/10.7554/eLife.79511.sa1
Author response https://doi.org/10.7554/eLife.79511.sa2

# Additional files

## Supplementary files

- Supplementary file 1. RNA seq for SAM synthase knockdown in basal conditions. Tabs A-C show *sams-3, sams-4, sams-5 (RNAi)* RNA seq data then Tabs D-F show WormCat gene enrichment. *sams-1* data is from *Ding et al., 2018*. Enriched categories from WormCat. Red color denoted categories with a p value of less than 0.01. NS is not significant, NV is no value, RGS is regulated gene set.

- Supplementary file 2. Statistics for survival curves. Each tab contains data for replicate experiments (R1, R2, R3). Statistical information from GraphPad Prism is also included.

- Supplementary file 3. CUT&TAG peaks for H3K4me3 from sams-1 and sams-4 animals in basal and heat shocked samples. Tabs A-F: Cut and Tag peaks from Control, *sams-1* and *sams-4* RNAi animals at 15 and 37 degrees determined by HOMER. Tabs G-I: Enriched categories from WormCat. Color denoted categories with a p value of less than 0.01 NS is not significant, NV is no value, RGS is regulated gene set.

- Supplementary file 4. Limited activation of heat shock response in *sams-4* RNAi animals. Tabs show RNA seq from control (A), *sams-1* (B) or *sams-4* (C) animals subjected to heat shock that was used for comparison with C&T data. Differential genes were identified using Deseq2 in DolphinNext. Data for control and *sams-1* RNAi animals is from *Ding et al., 2018*. WormCat batch output of two-fold regulated genes for Categories 1, 2, and 3 are in tabs (E-G). Highlighting denotes genes with significantly p values. NS is not significant, NV is no value, RGS is regulated gene set.

- MDAR checklist

## Data availability

Sequencing data have been deposited in GEO under accession code GSE223597.

The following previously published dataset was used:

| Author(s) | Year | Dataset title | Dataset URL | Database and Identifier |
| --- | --- | --- | --- | --- |
| Wei D, Daniel PH, Dilip KY, Adwait AG, Read P | 2018 | *C. elegans* stress-induced gene expression in low SAM or after histone-methytransferase RNAi | https://www.ncbi.nlm.nih.gov/geo/query/acc.cgi?acc=GSE121511 | NCBI Gene Expression Omnibus, GSE121511 |

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
