## [Editor Report]

The manuscript by Godbole et al. proposes a novel mechanism by which different S-adenosylmethionine (SAM) synthase enzymes exhibit specificity towards target sequences, establishing a layer of control over H3K4 trimethylation (H3K4me3). The authors demonstrate that the loss of two SAMs (*sams-1* and *sams-4*) differentially impacts stress response phenotypes, histone methylation, and gene expression profiles. This work suggests a role of enzyme provisioning in selecting specific targets for epigenetic modification.

---

## [Decision Letter]

**Decision letter after peer review:**

Thank you for submitting your article "S-adenosylmethionine synthases specify distinct H3K4me3 populations and gene expression patterns during heat stress" for consideration by *eLife*. Your article has been reviewed by 3 peer reviewers, one of whom is a member of our Board of Reviewing Editors, and the evaluation has been overseen by Jessica Tyler as the Senior Editor. The reviewers have opted to remain anonymous.

The reviewers have discussed their reviews with one another, and the Reviewing Editor has drafted this to help you prepare a revised submission. Please also refer to the full reviewers' comments for suggestions and questions regarding the interpretation of the results.

Essential revisions:

1) SAM levels were only measured in sams-4(RNAi), presumably not under the heat shock condition (which should be clearly stated in the manuscript). How were SAM levels altered by heat shock and the SAM synthetase mutants? The authors should thoroughly determine SAM levels in wild type, sams^-1^(lof), sams-4(RNAi), and sams1(lof)sams-4(RNAi) over their heat shock assay (e.g. before, during, and after the heat shock).

2) What is the survival phenotype for sams1(lof)sams-4(RNAi) to heat shock?

3) The authors make the point that sams-4 RNAi can also target sams-3, but for the most part, this is not addressed in the interpretation of the results of sams-4 RNAi data. It would be helpful for the authors to establish more firmly that the major target of sams-4 RNAi is really sams-4, not sams-3. The reviewers make several suggestions about possible experiments to do this but the authors may alternatively wish to address this point in a different fashion.

4) The authors report that gene expression changes are different for KD of sams^-1^ and sams-4. A useful way to show such a differential effect would be a PCA of RNAseq count data comparing control, sams^-1^, and sams-4 (with separate repeats) in the same figure. Such a figure would allow a comparison of the changes and location relative to dispersion of global gene expression in response to KD of SAMS.

5) The authors demonstrate that peaks for fbxa-59 and T27F6.8 are still impacted by heat stress in sams^-1^ but not in sams-4 RNAi animals and these targets are also differentially expressed. It would be interesting to confirm (RNAi?) if induction of either of these two genes is required e.g. for survival following heat shock?

6) Additional quality control analysis should be presented. E.g. H3K4me3 data should be compared with published ChIP-seq data. Also, if replicates were performed, the correlation between the replicates should be presented.

7) The reviewers noted favorably that the paper describes one of the first adaptations of CUT&TAG in *C. elegans* but also noted a lack of experimental detail making it difficult to fully evaluate the CUT&TAG data. The reviewers suggested a number of additions and controls in this context. Please address the points below:

a. It is unclear whether the authors performed any biological replicates for the CUT&TAG analysis.

b. What was the method of cell dissociation.

c. Antibody used for H3K4me3? (same as immunofluorescence?)

d. Please indicate internal controls? IgG, H3, etc.?

e. What modifications were made to the Henik_off_ Lab CUT&TAG protocol?

f. Please include a full method of library preparation for CUT&TAG?

g. What kind of peaks were called? Narrow? Broad? Combined?

h. Were true differential peaks called? I.e. peaks with a significant difference according to a statistical software in group A vs B? If so, include significance cutoffs and more details. If not, rephrase “differential peaks”.

i. How are the CUT&TAG data that are displayed normalized? To library size?

How were peaks annotated to genes?

j. What is being displayed on the y axis of all metaplots centered around the TSS? (e.g. Figure 2B)

k. How is the overlapping peak analysis done (i.e. how many basepairs must overlap to indicate an overlap of peaks?).

*Reviewer #1 (Recommendations for the authors):*

1) The authors make the point that sams-4 RNAi can also target sams-3, but for the most part, this is not addressed in the interpretation of the results of sams-4 RNAi data. In particular, the authors show the heat stress phenotype of sam-4 mutants and make a note on page 9, line 168 that "sams-4 depletion is the primary basis of the heat shock phenotypes and validate use of this RNAi strain to examine sams-4 function". However, the data for sams-4 RNAi is not shown for the heat shock phenotype, only the sams-4 mutant. It will be important for the authors to show both sams-4 RNAi and mutant data with a comparable phenotype in order to conclude that the two can be used interchangeably. Similarly, it would be helpful for the authors to include a heat shock assay of sams-3 RNAi alone so readers can assess whether sams-3 might really contribute to the heat shock phenotype or not. Since the authors at several times point out that the difference between sams^-1^ and sams-4 individual proteins is of interest, it would be helpful for the authors to establish more firmly that the major target of sams-4 RNAi is really sams-4, not sams-3.

2) The results of the heat stress survival assays in Figure S2 are puzzling, especially taken together with the immunostaining experiments in Figure S2 B-E. Since the loss of set-2 and set-16 increases the survival of both sams^-1^ and sams-4 mutants which have opposite heat stress survival phenotypes and opposite intestinal H3K4me3 staining phenotypes, it would almost seem more likely that the heat stress survival phenotype could be unrelated to H3K4me3. The authors should at a minimum discuss their interpretation of these results further in the text. Similarly, it is puzzling that neither set-2 or set-16 RNAi caused a decrease in H3K4me3 staining in sams^-1^ mutants under heat stress at all. If the authors mean to suggest that the enzymes can compensate for each other, it would be helpful to show a double RNAi of set-2 and set-16 and add some explanation in the text. Given that sams^-1^ mutants lack H3K4me3 in intestinal cells at 15C but gain it at 37C, they should require an H3K4 trimethylase enzyme, so if this is not the case, the authors will need to explain what they think the results indicate in more detail. Adding immunostaining experiments of sams-4 mutant could be helpful to interpret the overall results.

3) Overall, the results data from some of the experiments/methods (especially immunostaining, heat stress survival, and CUT&TAG) do not always fit well together. The authors chose excellent experiments to perform and did well to not over-interpret their results to try to force the results experiments to fit together, however, the end of each Results section and the discussion would benefit from increased discussion to reconcile the results together. For instance, pointing out that the results are unexpected but discussing possible reasons for the results would be beneficial for readers to understand the authors' interpretation of the results.

4) In general, the methods section needs to be more detailed. This is true for most sections, but details need to be added particularly for the CUT&TAG section, since the lack of detail here makes it difficult to interpret the data, and since CUT&TAG has rarely been used in *C. elegans*, and it is unclear whether the authors performed any biological replicates for the CUT&TAG analysis. The following details in specific should be added:

CUT&TAG:

Method of cell dissociation.

Antibody used for H3K4me3? (same as immunofluorescence?)

Any internal controls? IgG, H3, etc.?

Replicates?

What modifications were made to the Henik_off_ Lab CUT&TAG protocol?

Method of library preparation for CUT&TAG?

What kind of peaks were called? Narrow? Broad? Combined?

Were true differential peaks called? I.e. peaks with a significant difference according to a statistical software in group A vs B? If so, include significance cutoffs and more details. If not, rephrase "differential peaks".

How are the CUT&TAG data that are displayed normalized? To library size?

How were peaks annotated to genes?

What is being displayed on the y axis of all metaplots centered around the TSS? (e.g. Figure 2B)

How is the overlapping peak analysis done (i.e. how many basepairs must overlap to indicate an overlap of peaks?).

Other sections that need expansion.

Method or company used for creating CRISPR-tagged strains.

RNAi (method of making RNAi plates and growing bacteria, how much IPTG was used, etc.).

Lifespan methods and temperature of lifespan experiment.

RNA-seq (generation of heatmaps, etc.).

Immunofluorescence (add DAPI).

Methods for wormcat analysis.

Methods for mass spec.

Methods section that needs removal.

qPCR methods – no qPCR data are shown.

5) Additional quality control analysis should be presented. E.g. H3K4me3 data should be compared with published ChIP-seq data. Also, if replicates were performed, the correlation between the replicates should be presented.

6) The author identified a few factors (fbxa-59, T27F6.8, nhr-68) that showed both distinct H3K4me3 peaks and differential RNA expression between sams^-1^ and sams-4 mutants. The authors should test whether these factors could affect the physiological phenotypes (heat stress survival and H3K4me3 deposition in intestinal nuclei before and after heat shock).

7) Examining H3K4me3 levels only at TSS site may miss the potential difference between sams^-1^ and sams-4 that occurred at other genomic regions which may provide important information to address the differential heat shock survival phenotypes. This is particularly true for sams-4 dependent peaks (for example, Figure 2J), which showed a decrease in signal at the TSS. It is possible that sams-4 dependent peaks could be present in regions other than the TSS, and repeating the peak distribution analysis (as in Figure 2A) for these subsets of peaks could result in important insight about those peaks which show less TSS enrichment.

*Reviewer #2 (Recommendations for the authors):*

It is unclear how SAMS^-1^ and SAMS-4 differently affect the enrichment patterns of H3K4me3 during the heat stress response. Several key questions remain unresolved. First, SAM levels were only measured in sams-4(RNAi), presumably not under the heat shock condition (which should be clearly stated in the manuscript). How were SAM levels altered by heat shock and the SAM synthetase mutants? The authors should thoroughly determine SAM levels in wild type, sams^-1^(lof), sams-4(RNAi), and sams1(lof)sams-4(RNAi) over their heat shock assay (e.g. before, during, and after the heat shock). These results will be crucial for understanding whether SAM availability restricts H3K4 methylation under their experimental setting. Second, the author should test whether the response of H3K4me3 decrease was primary to SAM provision, but not due to a secondary effect like a result of the H3 deposition defect. H3 CUT&tag should be done as a control. Last, the author should explain why two SAM synthetases affect H3K4me3 differently? One of the enzymes might form a complex as found in yeasts? Or having distinct compartmentalized cellular localizations? Experiments addressing this point will be nice, but at least discussions should be expanded on this.

Other points:

What is the survival phenotype for sams1(lof)sams-4(RNAi) to heat shock?

Describe how the absolute concentration of SAM was determined in the method.

Figure S2G-J. I found the epistasis analysis a bit confusing and very inconclusive. If the survival defect of sams-4(RNAi) was due to SAM provision deficiency for H3K4me3, particularly for a subpopulation of H3K4me3 installed by a SAM-sensitive methyltransferase (MTase), deletion of that MTase will likely cause a similar survival defect. This is seen in the set-16(RNAi) mutant. Isn't this suggesting SET-16 being a SAM-sensitive MTase functionally important in the heat shock response? Further, I would think that SAM produced from SAMS^-1^ or SAMS-4 has broader usage in addition to histone methylation. It was thus unsurprising that the survival phenotype was worst in sams-4(RNAi).

It is not convincing that H3K4me3 alterations under heat shock were a direct response to the loss of SAMS-4. In addition to determining SAM levels as mentioned above, how about other modifications, and how H3K4me3 was temporally altered during this heat shock process?

For the pathways and genes that were differentially affected by sams^-1^(RNAi) and sams-4(RNAi), how did they functionally affect the heat shock response? None of these was tested or validated in this study.

Pg. 11, Line 208, missing a reference or a figure callout.

Pg. 12, Line 241, the figure callout here, Figure 2B, was not right.

Pg. 13, Line 255, missing a figure callout.

*Reviewer #3 (Recommendations for the authors):*

1) The method section appears somewhat light on detail in places:

a. As somebody who does not extensively work in this area, I found some of the technical aspects hard to follow and even harder to evaluate. Some more in-depth descriptions of the novel methods, e.g. the CUT&TAG approach, potentially with a supplementary diagram and validation data would make this aspect easier to understand – and certainly evaluate – for a wider audience.

b. Given that key conclusions and claims are based on sophisticated bioinformatics analysis, it would be important that all raw data is made available in a format that would allow others to replicate the full analysis. Scripts and parameter files should also be made available for the same reason.

c. Some additional information regarding software and tools would be welcome. e.g. what tools/packages were used for each of the statistical analyses and visualization of data and to generate final figures?

2) The authors report that gene expression changes are different for KD of sams^-1^ and sams-4. A useful way to show such a differential effect would be a PCA of RNAseq count data comparing control, sams^-1^ and sams-4 (with separate repeats) in the same figure. Such a figure would allow a comparison of the changes and location relative to dispersion of global gene expression in response to KD of SAMS.

3) How did total levels of SAM change with each of the interventions (KD, mutation of either SAMS?). It would be helpful to have an idea of how dependent the SAM pool is on these enzymes.

4) The authors demonstrate that peaks for fbxa-59 and T27F6.8 are still impacted by heat stress in sams^-1^ but not in sams-4 RNAi animals and these targets are also differentially expressed. I was excited to see the analysis of downstream targets pinpointing some specific targets. It would be interesting to confirm if induction of either of these two genes are required e.g. for survival following heat shock? Even if the impact of these targets was limited, such data would still be valuable.

---

## [Author Response]

Essential revisions:1) SAM levels were only measured in sams-4(RNAi), presumably not under the heat shock condition (which should be clearly stated in the manuscript). How were SAM levels altered by heat shock and the SAM synthetase mutants? The authors should thoroughly determine SAM levels in wild type, *sams-1(lof)*, *sams-4(RNAi)*, and *sams-1(lof);sams-4(RNAi)* over their heat shock assay (e.g. before, during, and after the heat shock).

In this study, as the reviewer states, we only reported SAM levels from *sams-4(RNAi)* animals, as we had determined SAM levels from *sams-1(RNAi)* animals in two previous studies (Walker, et al. Cell 2011 and Ding, et al. Cell Met 2015). We apologize this was not clearer in the text. However, we agree with the value of a direct comparison of SAM levels between the synthases, as well as a measurement under heat stress. We performed metabolomics on *sams-1* and *sams-4* animals in basal conditions and directly after a 2 hour heat shock. We did not include a rest or recovery period before collecting samples for IF, RNA seq or Cut&Tag, therefore our metabolomics are consistent with during/after heat shock and our other assays.

The metabolomics were informative (Figure 1 —figure supplement 3F-H), and we appreciate the reviewer’s suggestion. First, we were able to show side by side that SAM was decreased similarly after *sams-1* and *sams-4(RNAi)* in basal conditions. Second, we saw that SAM was increased in *sams-1(RNAi)*, but not *sams-4(RNAi)* animals after heat shock. This is an important result, as it is consistent with the H3K4me3 appearing in *sams-1(RNAi)* or *(lof)* intestinal nuclei after heat shock. The *sams-1(lof); sams-4(RNAi)* animals did not have viability to obtain sufficient populations for metabolomics and rescue of larval development with dietary choline, as we did in our IF assays, is likely to confound the metabolomics. We acknowledge this experiment would be informative and have added a statement to the limitations section.

2) What is the survival phenotype for sams1(lof)sams-4(RNAi) to heat shock?

We agree this is an important experiment and appreciate the reviewer’s suggestion.

Loss of multiple SAM synthases causes larval lethality (Twobin, Cell 2012) and *sams1(lof); sams-4(RNAi)* animals are likely to also have reduction in *sams-3* due to cotargeting with the *sams-4* RNAi. For our IF assays, we circumvented the lethality by rescuing PC production with dietary choline until mid-way through larval development. This approach provided sufficient sample sizes for IF studies. However, for survival assays, we took a different approach and used RNAi to knockdown *sams-1* in *sams-4* animals, removing potential effects from off target RNAi effects on *sams-3*. First, we confirmed that H3K4me3 dynamics in *sams-1(RNAi); sams-4(ok3315)* animals were similar *to sams-1(lof); sams-4(RNAi)* (Figure 1 —figure supplement 3D), then performed the survival assays (Figure 1 —figure supplement 3E). We found deletion of *sams-4*

reduced the survival in *sams-1* animals, demonstrating the importance of *sams-4* for the advantage provided by loss of *sams^-1^*.

3) The authors make the point that sams-4 RNAi can also target sams-3, but for the most part, this is not addressed in the interpretation of the results of sams-4 RNAi data. It would be helpful for the authors to establish more firmly that the major target of sams-4 RNAi is really sams-4, not sams-3. The reviewers make several suggestions about possible experiments to do this but the authors may alternatively wish to address this point in a different fashion.

We agree that the co-targeting of *sams-3* and *sams-4* in RNAi assays introduced unnecessary confusion and have taken multiple approaches to clarify this in our revision. First, we asked if heat shock phenotypes were specific to *sams-3* or *sams-4*. Using deletion alleles, we established that *sams-4(ok3315)* and not *sams-3(2932)* was required for survival after heat shock. Furthermore, the IF experiments described above, measuring H3K4me3 levels in *sams^-1^(RNAi) sams-4(ok3315)* animals demonstrate that loss of *sams-4* is sufficient to drive the phenotype. However, differences in growth timing of *sams-1(lof)* and wild type animals make it less desirable to use for large scale assays such as RNAseq or the Cut&Tag assays. Therefore, we also sought to clarify co-targeting effects of *sams-4* RNAi to more accurately describe these assays. We used either RNAi to *sams-3* or *sams-4* to determine knockdown of endogenously tagged *sams-3*::mKate or *sams-4*::GFP animals (Figure 1 —figure supplement1C) and noted that while RNAi to *sams-3* appeared to affect *sams-3*::mKate and *sams-4*::GFP at similar, robust levels, RNAi to *sams-4* had a greater effect on *sams-4*::GFP, although there were partial effects on *sams-3*::mKate (Figure 1 —figure supplement1C). We also included SAM levels in *sams-3(RNAi)* animals (Figure 1 figure supplement1D).

Taken together, we conclude that the heat shock phenotypes are linked to *sams-4*, rather than *sams-3*, but that *sams-4(RNAi)* can also affect *sams-3*. Therefore, we have updated the text and figure legends to clarify this point.

4) The authors report that gene expression changes are different for KD of sams^-1^ and sams-4. A useful way to show such a differential effect would be a PCA of RNAseq count data comparing control, sams^-1^, and sams-4 (with separate repeats) in the same figure. Such a figure would allow a comparison of the changes and location relative to dispersion of global gene expression in response to KD of SAMS.

We appreciate this suggestion to visualize and provide more information of our analysis and have added PCA of the basal RNA seq (Figure 1—figure supplement 2A) and after heat shock (Figure 4—figure supplement 1A).

5) The authors demonstrate that peaks for fbxa-59 and T27F6.8 are still impacted by heat stress in sams^-1^ but not in sams-4 RNAi animals and these targets are also differentially expressed. It would be interesting to confirm (RNAi?) if induction of either of these two genes is required e.g. for survival following heat shock?

We understand the additional insights that could be made if “target” genes could be identified. We also appreciate the reviewer’s understanding that the strength of our study lies in the demonstration of genome-wide distinctions in *sams-1* and *sams-4*dependent H3K4me3 patterns and that phenotypes such as survival after heat shock may require multiple genes. We picked three genes with altered H3K4me3 peaks to test for effects on survival. First, as the reviewer’s suggested, we compared *fbxa-59* and T27F6.8 RNAi to control and found a slight, but statistically significant effect limiting survival with *fbxa-59*, suggesting it could be part of a broader program (Figure 4—figure supplement 1C). We found no effects after RNAi of T27F6.8 (Figure 4—figure supplement 1B). (Although it is preferred to name *C. elegans* genes discussed in publications, we did not request a name for T27F6.8, as it did not produce a phenotype.) Second, we noted that H3K4me3 peaks for *nhr-68* were lower in *sams-1* animals in basal conditions and that its expression was reduced after heat shock. We hypothesized that expression from an autologous promoter refractory to *sams-1* dependent effects on H3K4me3 might change survival. We obtained strain expressing *nhr-68* from a *ges-1* promoter from the Walhout lab (Blucha, et al. Cell Reports 2019). *ges-1* is expressed specifically in the intestine, and we noted no changes in H3K4me3 on its promoter (Figure 6 —figure supplement 1A). Expression of *nhr-68* from an H3K4me3-indepent promoter diminished survival after heat shock, suggesting its regulation may be an important part of the pro-survival program (Fig6E).

6) Additional quality control analysis should be presented. E.g. H3K4me3 data should be compared with published ChIP-seq data. Also, if replicates were performed, the correlation between the replicates should be presented.

We have added Pearson correlation plots comparing replicates and to ChIP seq data (Figure 3—figure supplement 1A, B). Although there are multiple ChIP seq datasets, each had significant differences in biological context from our assays, either because the assays were performed in L3 larvae (modEncode) or germline animals (Pu, et al.). We did find the strongest correlation with the Wan et al. data, which was from wild type adults, as in our assays. However, the animals in the Wan et al. data were fed OP50 rather than the HT115 bacteria, which could contribute to some differences.

7) The reviewers noted favorably that the paper describes one of the first adaptations of CUT&TAG in *C. elegans* but also noted a lack of experimental detail making it difficult to fully evaluate the CUT&TAG data. The reviewers suggested a number of additions and controls in this context. Please address the points below:

We appreciate the reviewer’s suggestions regarding improving and highlighting our Cut&Tag studies. Each point will be answered individually, although some of our changes may be relevant to multiple points.

a. It is unclear whether the authors performed any biological replicates for the CUT&TAG analysis.

The assays were performed in duplicate. We have updated the text to clarify this as well as providing a correlation plot of the replicates (Figure 3—figure supplement 1A, B).

b. What was the method of cell dissociation.

*C. elegans* were dissociated with dounce homogenization. We have updated the methods section to add this and other details to our protocol.

c. Antibody used for H3K4me3? (same as immunofluorescence?)

We have updated the methods to clarify that the same antibody was used for IF and Cut&Tag (Cell Signaling, C42D8).

d. Please indicate internal controls? IgG, H3, etc.?

Cut&Tag depends on antibody binding and localization of the transposase to release DNA for library production and sequencing. H3 controls may be less informative as cutting would occur at every histone. Therefore, we used a sample with no antibody as a control. Our library sizes for samples with H3K4me3 antibodies ranged from 5x10^4^ to 7x10^5^. No antibody libraries ranged from 1x10^3^-1x10^4^, providing very low reads as expected for an internal negative control. We have added a browser track showing the no antibody control for *pcaf-1*, a positive control for promoter localized H3K4me3 used previously by our lab (Ding, et al. Cell Metab 2015) and others (Xiao, et al. PNAS 2011) as Figure 3—figure supplement 1C.

e. What modifications were made to the Henikoff Lab CUT&TAG protocol?f. Please include a full method of library preparation for CUT&TAG?

We have updated the methods to more completely describe our protocol.

g. What kind of peaks were called? Narrow? Broad? Combined?

The HOMER suite uses *-style histone* to specify broad peaks. We have clarified this in the Data Analysis section.

h. Were true differential peaks called? I.e. peaks with a significant difference according to a statistical software in group A vs B? If so, include significance cutoffs and more details. If not, rephrase “differential peaks”.

We used the default commands in HOMER (getDifferentialPeaks) to define differential peaks. This has been included in the Data analysis methods.

i. How are the CUT&TAG data that are displayed normalized? To library size?How were peaks annotated to genes?

We normalized Cut&Tag data to library size and peaks were annotated by HOMER with the “annotate peaks command”. This has been added to the Data Analysis section.

j. What is being displayed on the y axis of all metaplots centered around the TSS? (e.g. Figure 2B)

The Y axis shows Peaks per base pair of gene. We have updated the figure legends to clarify this point.

k. How is the overlapping peak analysis done (i.e. how many basepairs must overlap to indicate an overlap of peaks?).

We used the findOverlapsOfPeaks command in ChipSeqAnno (Zhu, et al. 2010) with a max gap of 1000 basepairs to determine peak overlap. This has been updated in the methods.